# Universal Image Immunization against Diffusion-based Image Editing via Semantic Injection

## Abstract

Recent advances in diffusion models have enabled powerful image editing capabilities guided by natural language prompts, unlocking new creative possibilities. However, they introduce significant ethical and legal risks, such as deepfakes and unauthorized use of copyrighted visual content. To address these risks, image immunization has emerged as a promising defense against AI-driven semantic manipulation. Yet, most existing approaches rely on image-specific adversarial perturbations that require individual optimization for each image, thereby limiting scalability and practicality. In this paper, we propose the first universal image immunization framework that generates a single, broadly applicable adversarial perturbation specifically designed for diffusion-based editing pipelines. Inspired by universal adversarial perturbation (UAP) techniques used in targeted attacks, our method generates a UAP that embeds a semantic target into images to be protected. Simultaneously, it suppresses original content to effectively misdirect the model's attention during editing. As a result, our approach effectively blocks malicious editing attempts by overwriting the original semantic content in the image via the UAP. Moreover, our method operates effectively even in data-free settings without requiring access to training data or domain knowledge, further enhancing its practicality and broad applicability in real-world scenarios. Extensive experiments show that our method, as the first universal immunization approach, significantly outperforms several baselines in the UAP setting. In addition, despite the inherent difficulty of universal perturbations, our method also achieves performance on par with image-specific methods under a more restricted perturbation budget, while also exhibiting strong black-box transferability across different diffusion models.

## 1 Introduction

Diffusion models have emerged as a dominant paradigm in image synthesis, generating high-fidelity, semantically rich images by iteratively denoising Gaussian noise through a learned reverse process (Dhariwal & Nichol, 2021; Ho et al., 2020; Song et al., 2021). Conditional extensions guided by text prompts, spatial masks, or reference images via cross-attention further enable fine-grained, user-controllable generation, and editing (Rombach et al., 2022; Brooks et al., 2023; Zhang et al., 2023b). While diffusion models enable creative and interactive applications, their powerful user-guided generation capabilities also raise serious concerns about misuse, such as imperceptible malicious edits, identity spoofing, or unauthorized replication of copyrighted content. This highlights the need for robust defense mechanisms to mitigate harmful edits generated by diffusion models.

To prevent unauthorized image editing, adversarial perturbation-based immunization methods (Yeh et al., 2021; Ruiz et al., 2023; Salman et al., 2023; Lo et al., 2024; Choi et al., 2025) inject carefully crafted, imperceptible perturbations into source images to disrupt potential manipulations. Early works (Ruiz et al., 2023; Aneja et al., 2022; Yeh et al., 2021) in this direction primarily focused on defending against manipulation techniques based on Generative Adversarial Networks (GANs). With the growing adoption of diffusion models, recent works (Salman et al., 2023; Lo et al., 2024; Choi et al., 2025; Jeon et al., 2025) have explored immunization strategies against malicious diffusion-based editing. For text-guided image editing, PhotoGuard (Salman et al., 2023) was the first to introduce encoder- and diffusion-level attacks by crafting imperceptible perturbations, targeting diffusion models. Semantic Attack (Lo et al., 2024) enhances robustness by localizing perturbations

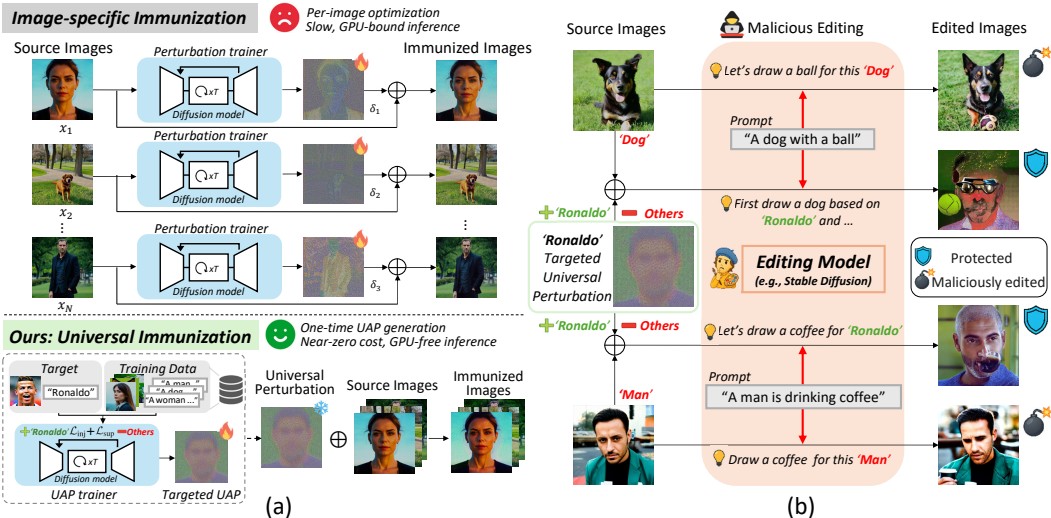

Figure 1: Illustrations of our universal immunization approach and the motivation behind semantic injection. (a) Unlike image-specific approaches (top) requiring costly per-image processing, our universal immunization method (bottom) employs a single, pre-computed UAP to safeguard images without any inference-time overhead. (b) Injecting the target content into source images via a UAP misleads the editing model, causing it to lose the original content and resulting in failed edits. Note that perturbations are scaled for better visualization of details.

to semantically critical regions, selectively disrupting text-to-image alignment. More recently, DiffusionGuard (Choi et al., 2025) and AdvPaint (Jeon et al., 2025) have been proposed as defense frameworks that optimize adversarial perturbations specifically tailored to diffusion-based inpainting. Despite their effectiveness, these methods share a key limitation: they rely on image-specific perturbations that must be optimized independently for each input, thereby limiting scalability and hindering practical deployment in real-world applications.

In this paper, we propose a novel method for generating universal adversarial perturbations (UAPs) to immunize images against diffusion-based image editing. As shown in Figure 1(a), unlike the image-specific methods that rely on computationally intensive, per-image GPU optimization, our approach offers a universal, plug-and-play solution that eliminates test-time computation, making it readily applicable for large-scale real-world deployment. Inspired by targeted UAPs in classification (Zhang et al., 2020), our method embeds intended target semantics into source images to overwrite their original meaning during editing. To effectively inject target semantics into a UAP, we introduce two loss functions: a *target semantic injection* loss, which encourages the embedding of target content into the perturbation, and a *source semantic suppression* loss, which reduces the influence of the source image's content. Together, these objectives steer the perturbed image toward the target semantics, prompting diffusion-based editors to act on the injected signal instead of the source content, thereby mitigating unauthorized or malicious edits.

Our motivation for semantic injection is illustrated in Figure 1(b). Our UAP embeds the semantics of the target content, *'Ronaldo'*, into the source image, making the editing model misinterpret the original content and generate an output unrelated to it. Additionally, the UAP jointly suppresses the original semantics of the source image, further reinforcing the dominance of the target and guiding the editing process toward target-driven generation. The effectiveness of our method is further demonstrated in Figure 2. As shown in Figure 2(b), without immunization, the source image activates attention for the actual contents presented in the images, such as *'cow'* and *'people'*. In contrast, the attention map of the immunized image (Figure 2(c)) is sharply focused on the intended target, *'Ronaldo'*, closely resembling the attention pattern in the target image (Figure 2(a)), while failing to attend to the original semantic content. This indicates that our UAP aligns the model's focus with the desired target semantic content and effectively overrides the original semantic representation.

Extensive experiments demonstrate that our method consistently and substantially outperforms all baselines across both quantitative metrics and qualitative analyses. Moreover, it exhibits strong black-box transferability, maintaining high performance across diverse diffusion-based editing models without model-specific tuning. Notably, our approach even surpasses several state-of-the-art image-

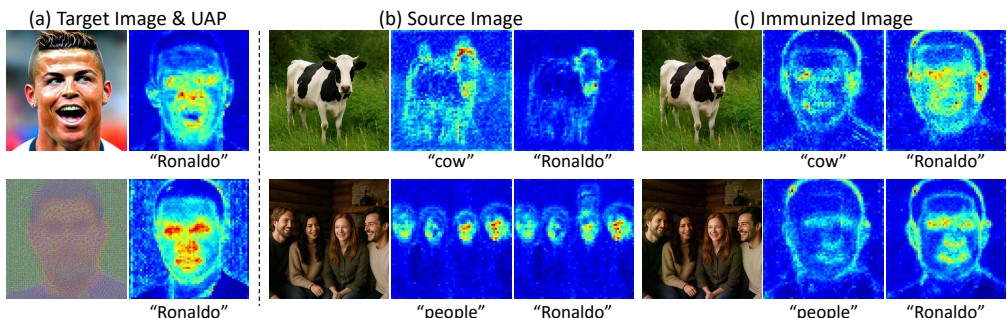

Figure 2: Visualization of cross-attention maps from the diffusion model under different conditions. (a) Target images generated by Stable Diffusion (Rombach et al., 2022) conditioned on the prompt 'Ronaldo', along with the generated corresponding UAP. (b) Source images show strong attention to their own content, but since they do not intrinsically contain the target semantics ('Ronaldo'), they do not produce any target-aligned attention. (c) Immunized images, perturbed by the UAP, fail to focus on the original prompt and instead exhibit strong attention to the target concept 'Ronaldo'. Each attention map is annotated with the prompt used for conditioning.

specific methods while reducing inference cost to nearly zero. Finally, it remains effective in data-free scenarios, where no real-world training data or prior domain knowledge is available, underscoring its practicality for real-world deployment.

The main contributions can be summarized as follows:

- We propose a novel universal immunization strategy that uses targeted universal adversarial perturbations (UAPs) to defend against malicious edits in diffusion-based models, without runtime overhead. To the best of our knowledge, this is the first work to achieve image immunization via a single, general-purpose perturbation.

- We introduce two loss functions for training our UAP: the *target semantic injection* loss encourages the injection of intended target semantics, and the *source semantic suppression* loss suppresses original concepts present in the training set, enabling effective immunization on unseen inputs.

- Our universal approach not only consistently surpasses universal baselines across diverse diffusion models, including our data-free variant, but also achieves competitive performance with image-specific immunization methods despite the inherent difficulty of the universal setting, demonstrating strong black-box generalization and high test-time efficiency.

## 2 RELATED WORK

**Text-Conditioned Diffusion Models.** Text-to-image (T2I) diffusion models generate images conditioned on text prompts by learning to reverse a gradual noising process. Recent works such as DALL·E 2 (Ramesh et al., 2022), Imagen (Saharia et al., 2022), and Stable Diffusion (Rombach et al., 2022) have demonstrated remarkable performance by leveraging powerful language models and cross-attention mechanisms. Building upon the success of T2I models, several approaches have been developed to enable text-guided image editing. DiffEdit (Couairon et al., 2022) introduces a mask-based editing framework that identifies editable regions in the image, while Imagic (Kawar et al., 2023) fine-tunes diffusion models on individual images to better preserve identity during editing. InstructPix2Pix (Brooks et al., 2023) further advances this line of the works by training on instruction-edit pairs, allowing for generalizable and intuitive editing based on natural language instructions. In addition, Stable Diffusion (Rombach et al., 2022) provides native support for text-guided image editing tasks such as img2img and inpainting, enabling users to modify existing images in a semantically coherent way based on textual prompts. While these models exhibit remarkable performance, their growing risk of malicious use raises concerns about content authenticity and motivates research into image protection methods that enable safe and controllable editing.

**Image Immunization.** Image immunization has emerged as a promising solution, aiming to proactively disrupt the generative process of diffusion models and safeguard visual content from unauthorized manipulation. Early works (Yeh et al., 2021; Ruiz et al., 2023; Aneja et al., 2022) on image immunization primarily focused on GAN-based image manipulation techniques. More

recently, attention has shifted toward immunizing against diffusion-based editing (Salman et al., 2023; Lo et al., 2024; Choi et al., 2025; Jeon et al., 2025). PhotoGuard (Salman et al., 2023) first introduced an image immunization strategy tailored to diffusion models, aiming to prevent unauthorized edits. EditShield (Chen et al., 2024) maximizes the latent distance between perturbed and EOT-averaged source images to enhance robustness against instruction-guided image editing. Semantic Attack (Lo et al., 2024) extended this idea by leveraging attention maps to localize and suppress alignment in specific image regions. Pixel is Not a Barrier (Shih et al., 2025) further demonstrated that PDM exhibits greater robustness than LDM and propose a corresponding immunization method tailored to this property. To prevent malicious image inpainting, DiffusionGuard (Choi et al., 2025) perturbs the early denoising steps with robust adversarial noise via mask augmentation, while AdvPaint (Jeon et al., 2025) disrupts the attention mechanism within the U-Net architecture to suppress text-to-image alignment. While these works show promising results, they commonly rely on image-specific perturbations, which require additional costs for each image, limiting their scalability and practicality in real-world settings.

**Universal Adversarial Perturbation.** Universal Adversarial Perturbations (UAPs), first introduced in (Moosavi-Dezfooli et al., 2017), aim to generate a single, image-agnostic perturbation capable of fooling a wide range of inputs. UAPs can be crafted in either data-dependent (Liu et al., 2023a; Poursaeed et al., 2018; Shafahi et al., 2020) or data-free settings (Zhang et al., 2021; Mopuri et al., 2018; Liu et al., 2023b; Lee et al., 2025). Data-dependent methods rely on representative datasets to optimize perturbations across samples, while data-free methods use synthetic inputs, such as Gaussian noise or jigsawed images, making them suitable for scenarios where access to training data or the target domain is restricted. Yet, UAPs remain less effective than image-specific perturbations in classification, because the challenge of deceiving all images simultaneously inevitably limits their performance. Furthermore, the development of universal perturbations for generative models, particularly diffusion models, remains largely underexplored.

## 3 BACKGROUND AND MOTIVATION

**Targeted Universal Adversarial Perturbation (UAP).** UAPs have been explored in the context of targeted attacks (Zhang et al., 2020), demonstrating that a single, input-agnostic perturbation can effectively drive diverse inputs toward a specific target label $y_{\text{tar}}$. The objective is defined as:

$$\delta^* = \arg\min_{\delta} \ \mathbb{E}_{x \sim \mathcal{D}} \left[ \mathcal{L}(f(x + \delta), y_{\text{tar}}) \right], \quad \text{s.t.} \quad \|\delta\|_{\infty} \leq \epsilon, \tag{1}$$

where $\mathcal{D}$ is the data distribution, $x$ is an input image sampled from $\mathcal{D}$, $f$ is the classifier, $\mathcal{L}$ is the loss function (*e.g.*, cross-entropy), $\delta$ is the UAP to be optimized, and $\epsilon$ is the perturbation budget. By leveraging the diverse semantics of the data to encode target-aligned features into a UAP, costly per-instance optimization can be avoided.

Motivated by this, we extend targeted UAPs to immunize images against diffusion-based editing by embedding target semantics into a single, universal perturbation. Unlike prior image-specific approaches (Salman et al., 2023; Lo et al., 2024) that rely on slow per-instance optimization, our method trains a UAP that generalizes across inputs, enabling efficient and scalable protection against diffusion-based editing.

**Text-Image Cross-Attention in Diffusion Models.** Text-to-image diffusion models (Ramesh et al., 2022; Saharia et al., 2022; Rombach et al., 2022) generate images by progressively denoising latent variables while conditioned on textual inputs. Their training follows the standard diffusion objective:

$$\mathcal{L}(\theta) = \mathbb{E}_{x_0, t, \epsilon, c} \left[ \|\epsilon_{\theta}(x_t, k, t) - \epsilon\|_2^2 \right], \tag{2}$$

where the noise prediction network $\epsilon_{\theta}$ learns to denoise a wide variety of images while conditioning on diverse text prompts $t$. To achieve this, the U-Net backbone incorporates cross-attention layers at multiple spatial resolutions, allowing textual embeddings to influence the latent representation at each timestep $k$. Let $z \in \mathbb{R}^{N \times d}$ denote latent features and $t \in \mathbb{R}^{M \times d}$ the text embeddings. The cross-attention output is computed as:

$$\text{CA}(z, t) = \text{softmax}\left( \frac{Q(z)K(t)^{\top}}{\sqrt{d_k}} \right) V(t), \tag{3}$$

where $Q, K, V$ are learned projections. This mechanism enables each spatial location in the latent image to selectively attend to semantically relevant components of the text prompt.

Recent works (Lo et al., 2024; Jeon et al., 2025) have exploited cross-attention for image-specific immunization by manipulating attention maps (Lo et al., 2024) or altering intermediate representations, such as query embeddings (Jeon et al., 2025). However, they often lack a direct and scalable intervention for controlling target semantics within diffusion models. Distinct from previous works, our semantic injection approach guides diffusion models to interpret the immunized image itself as containing the target semantics. To accomplish this, the proposed losses operate directly on the cross-attention outputs. A more detailed theoretical justification is provided in Section 4.4.

# 4 METHOD

In this section, we present our universal immunization approach for disrupting diffusion-based image editing. Our method injects intended target semantics into images while suppressing the original content, and remains effective even in a data-free setting, enabling practical and scalable protection.

## 4.1 UNIVERSAL IMAGE IMMUNIZATION VIA SEMANTIC INJECTION

Previous image immunization methods primarily focus on image-specific approaches (Salman et al., 2023; Lo et al., 2024; Choi et al., 2025; Jeon et al., 2025), where a distinct adversarial perturbation is generated for each source image. While effective, these methods incur significant computational overhead to craft each immunized image, as illustrated in Figure 1(a). To address this, we propose a universal immunization method that learns a universal adversarial perturbation (UAP) through one-time training, which can then be effortlessly applied to any input at inference.

Inspired by targeted UAPs in classification (Zhang et al., 2020), our approach aims to overwrite the original semantics of source images during editing by embedding intended target concepts through an adversarial perturbation. To this end, we introduce two loss terms for training a single, general-purpose perturbation: the *target semantic injection loss* $\mathcal{L}_{\text{inj}}$, which encourages alignment with the intended target semantics, and the *source semantic suppression loss* $\mathcal{L}_{\text{sup}}$, which discourages retention of the original content. The overall optimization objective for training UAPs in our universal immunization framework, based on the two proposed loss terms, $\mathcal{L}_{\text{inj}}$ and $\mathcal{L}_{\text{sup}}$, is defined as follows:

$$\delta^* = \arg\min_{\delta} \ \mathbb{E}_{(x,t)\sim\mathcal{D}_p} \left[ \mathcal{L}_{\text{inj}} + \mathcal{L}_{\text{sup}} \right], \quad \text{s.t.} \quad \|\delta\|_{\infty} \leq \epsilon, \tag{4}$$

where $(x,t) \sim \mathcal{D}_p$ denotes an input image and its corresponding text embeddings from the image-prompt pair data distribution. Each loss component is described in detail below.

**Target Semantic Injection.** Our primary objective is to train a UAP that injects target semantics into source images within a diffusion-based image editing framework. To achieve this, we encourage the cross-attention responses of immunized images—perturbed by the UAP and conditioned on a target prompt—to closely match those of genuine target images guided by the same prompt. To ensure semantic alignment across multiple spatial feature levels, we aggregate cross-attention outputs from all layers of U-Net. Formally, the proposed *target semantic injection loss* is defined as follows:

$$\mathcal{L}_{\text{inj}} = \sum_{\ell=1}^{L} \left\| \text{CA}(\Phi^{\ell}(\mathcal{E}(x+\delta)), \ t_{\text{tar}}) - \text{CA}(\Phi^{\ell}(\mathcal{E}(x_{\text{tar}})), \ t_{\text{tar}}) \right\|_2^2, \tag{5}$$

where CA denotes the cross-attention output described in Eq. (3), $\Phi^{\ell}(\cdot)$ represents an intermediate feature map after $\ell$-th intermediate block of the denoising U-Net, $L$ is the number of intermediate blocks in the U-Net, $\mathcal{E}$ is an encoder of a variational auto-encoder (VAE) (Diederik P. Kingma, 2014; Van Den Oord et al., 2017). Here, $x_{tar}$ and $t_{tar}$ denote a target image and CLIP (Radford et al., 2021) text embedding of the target prompt (*e.g.*, 'Ronaldo', 'tiger'), respectively. The resulting UAP is trained to align the internal attention outputs of diverse source images with those of the target, effectively injecting target semantics in a consistent and image-agnostic manner. Even though the perturbed image does not perfectly resemble the target, the disruption of the original semantics is typically sufficient to prevent faithful content preservation during malicious editing.

**Source Semantic Suppression.** To further misalign the immunized image from its original semantics, we introduce a *source semantic suppression loss*. Inspired by prior work on adversarial attacks (Zhang et al., 2022), which improves attack effectiveness by minimizing the cross-entropy with the target label while maximizing it for non-target labels, we similarly aim to suppress source

semantics in diffusion models. Specifically, we maximize the discrepancy between the cross-attention outputs of the original image and its UAP-perturbed counterpart. The loss is defined as:

$$\mathcal{L}_{\text{sup}} = -\sum_{\ell=1}^{L} \left\| \text{CA}(\Phi^\ell(\mathcal{E}(x+\delta)),\, t) - \text{CA}(\Phi^\ell(\mathcal{E}(x)),\, t) \right\|_2^2. \tag{6}$$

The loss $\mathcal{L}_{\text{sup}}$ is jointly optimized with the target semantic injection loss $\mathcal{L}_{\text{inj}}$, enabling stronger targeted attacks by simultaneously erasing source semantics and reinforcing the injected target semantics. Specifically, while $\mathcal{L}_{\text{inj}}$ encourages alignment of the adversarial image with the target's semantics, $\mathcal{L}_{\text{sup}}$ drives its deviation from the original source semantics.

### 4.2 EXTENSION TO DATA-FREE SETTING

Our target semantic injection loss is designed to overwrite an image's original semantics with an arbitrary target, regardless of the input, making it effective even in data-free settings. While training with real data can enhance the consistency and strength of the injected semantics, we show that our method remains reliable without access to any real images. Following prior data-free universal attack methods (Lee et al., 2025; Liu et al., 2023b), we sample synthetic inputs $x_r$ from a random prior distribution $\mathcal{D}_r$ (*e.g.*, Gaussian noise or jigsaw puzzles) and treat them as training data. We then optimize UAP $\delta$ using only the target semantic injection loss in Eq. (5). This enables our data-free immunization approach to generate a targeted UAP from a single target image, without requiring real-world training data, while effectively defending against malicious editing across diverse inputs. Additional details and visualizations of random prior synthesis are provided in Appendix 7.3.

### 4.3 OVERALL ALGORITHM

The overall training and testing procedures for our universal image immunization framework are presented in Algorithm 1 and Algorithm 2, respec-

---

**Algorithm 1** UAP Training

**Input:** Training image set $X$, its corresponding prompt token embedding set $T$, target image $x_{tar}$, target prompt token embedding $t_{tar}$, surrogate diffusion model $\mathcal{M}$, diffusion timestep set $K = \{k_1, k_2, ..., k_n\}$, training epoch $N$, attack stepsize $s$, constraint parameter $\epsilon$
**Output:** Universal adversarial perturbation $\delta$.
1: Initialize $\delta \sim \mathcal{U}(-\epsilon, \epsilon)$
2: **for** $i = 1, \ldots, N$ **do**
3:    **for** $x \in X$ and $t \in T$ **do**
4:      $grad\_total \leftarrow 0$
5:      $x_{adv} \leftarrow \text{clip}(x + \delta, 0, 1)$
6:      **for** $k$ in $K$ **do**
7:        Add the forward-process noise to both clean and immunized images
        $x^k \leftarrow x + \text{noise\_scheduler}(k)$
        $x_{adv}^k \leftarrow x_{adv} + \text{noise\_scheduler}(k)$
        $x_{tar}^k \leftarrow x_{tar} + \text{noise\_scheduler}(k)$
8:        Compute loss using Eq. (5) and (6)
        $\mathcal{L} \leftarrow \mathcal{L}_{\text{inj}} + \mathcal{L}_{\text{sup}}$
9:        Update the total gradient
        $grad\_total \leftarrow grad\_total + \nabla \mathcal{L}$
10:     **end for**
11:     Update and clip perturbation $\delta$
      $\delta \leftarrow \delta - s \cdot sign(grad\_total)$
      $\delta \leftarrow \min(\epsilon, \max(\delta, -\epsilon))$
12:    **end for**
13: **end for**
14: **return** $\delta$

---

**Algorithm 2** Test-time Immunization

**Input:** New image $x$, pre-computed UAP $\delta$
**Output:** immunized image $x_{adv}$.
1: $x_{adv} \leftarrow x + \delta$
2: **return** $x_{adv}$

---

tively. We adopt the optimization strategy of the image-specific immunization method, Semantic Attack (Lo et al., 2024), extending it to a universal setting by training a single perturbation across the entire dataset instead of per-test image optimization. Once trained, the universal perturbation $\delta$ can be directly applied to any new input image at test time as $x_{\text{new}} + \delta$, enabling fast and scalable immunization.

### 4.4 THEORETICAL JUSTIFICATION FOR UTILIZING CROSS-ATTENTION OUTPUTS

In latent-diffusion training, the denoiser $\epsilon_\theta(x_t, k, t)$ is guided by the conditioning prompt $t$, causing the cross-attention pathway to continuously steer latent features toward prompt-aligned semantics throughout the denoising trajectory. At layer $l$, a cross-attention block computes:

$$Q_l = h_k^{(l)} W_l^Q, \quad K_l = Y W_l^K, \quad V_l = Y W_l^V, \tag{7}$$

where $Y$ denotes the prompt embedding. The attention map $\mathcal{A}_l$ and the output $CA_l$ are given by:

$$\mathcal{A}_l = \text{softmax}(Q_l K_l^\top / \sqrt{d_k}), \quad CA_l = \mathcal{A}_l V_l, \tag{8}$$

and the latent update follows:

$$h_k^{(l+1)} = h_k^{(l)} + G_l(h_k^{(l)}, k) + W_l^{CA} CA_l, \tag{9}$$

Figure 3: Qualitative comparison with universal baselines and image-specific methods. 'Enc.', 'Emb.', and 'Map' represent the universal baselines, obtained by adapting image-specific methods: encoder attack (Salman et al., 2023), intermediate representation disruption (e.g., query/key/value embeddings) (Jeon et al., 2025), and attention map attack (Lo et al., 2024), respectively. 'EA', 'DA', and 'SA' indicate image-specific immunization methods: Encoder-, Decoder- (Salman et al., 2023), and Semantic-Attack (Lo et al., 2024). We use '*Ronaldo*' as the target prompt for UAP generation.

Table 1: White-box immunization performance compared to universal baselines on Stable Diffusion v1.5. *Ours* denotes the full method, while *Ours$_{DF}$* uses only the target semantic injection loss in a data-free setting. The best result is shown in bold; the second-best is underlined.

| Method | Clean | Universal Baselines | | | Ours$_{DF}$ | Ours |
|---|---|---|---|---|---|---|
| | | Encoder | Embedding | Map | | |
| PSNR ↓ | – | $16.55_{\pm 0.073}$ | $15.80_{\pm 0.042}$ | $16.16_{\pm 0.056}$ | $\underline{14.68}_{\pm 0.035}$ | $\mathbf{14.19}_{\pm 0.059}$ |
| SSIM ↓ | – | $0.482_{\pm 0.004}$ | $\underline{0.378}_{\pm 0.003}$ | $0.468_{\pm 0.004}$ | $\underline{0.378}_{\pm 0.002}$ | $\mathbf{0.332}_{\pm 0.003}$ |
| VIFp ↓ | – | $0.154_{\pm 0.002}$ | $0.117_{\pm 0.001}$ | $0.152_{\pm 0.002}$ | $\underline{0.106}_{\pm 0.001}$ | $\mathbf{0.082}_{\pm 0.001}$ |
| FSIM ↓ | – | $0.714_{\pm 0.005}$ | $0.689_{\pm 0.004}$ | $0.710_{\pm 0.004}$ | $\underline{0.666}_{\pm 0.002}$ | $\mathbf{0.642}_{\pm 0.003}$ |
| LPIPS ↑ | – | $0.452_{\pm 0.005}$ | $0.548_{\pm 0.003}$ | $0.465_{\pm 0.004}$ | $\underline{0.557}_{\pm 0.002}$ | $\mathbf{0.606}_{\pm 0.003}$ |
| Feat. Sim. (C) ↓ | 0.744 | $0.708_{\pm 0.004}$ | $0.696_{\pm 0.005}$ | $0.704_{\pm 0.001}$ | $\underline{0.685}_{\pm 0.003}$ | $\mathbf{0.673}_{\pm 0.002}$ |
| Feat. Sim. (D) ↓ | 0.534 | $0.438_{\pm 0.004}$ | $0.436_{\pm 0.004}$ | $0.424_{\pm 0.003}$ | $\underline{0.376}_{\pm 0.003}$ | $\mathbf{0.345}_{\pm 0.004}$ |

where $G_l$ collects non–cross-attention operations (*e.g.*, convolution or self-attention) and $W_l^O$ is the output projection.

Our goal is to inject target semantics and suppress original semantics via conditioning prompts within the denoising pipeline. Since cross-attention is the primary pathway through which prompt information modulates the latent representation, it naturally serves as an effective point of intervention. To this end, we manipulate the cross-attention output $CA_l$ rather than the attention map $\mathcal{A}_l$, because $CA_l$ directly encodes semantic updates applied to the latent Eq. (9), whereas $\mathcal{A}_l$ only specifies token selection and does not contain semantic content. Moreover, $\mathcal{A}_l$ depends only on $Q_l$ and $K_l$ and remains unchanged under any invertible linear transformation applied to $V_l$, and therefore cannot uniquely represent the conditioning effect.

Thus, modifying $\mathcal{A}_l$ would alter only *where* to attend, not *what* semantics are injected. Accordingly, our Target Semantic Injection and Source Semantic Suppression losses are formulated on $CA_l$ to achieve reliable semantic control during denoising. This is further validated in Table 21 in the Appendix, showing that operating on $CA_l$ yields noticeably stronger semantic effects compared to manipulating attention maps $\mathcal{A}_l$.

## 5 EXPERIMENTS

### 5.1 EXPERIMENTAL SETUP

Since our work presents the first universal immunization approach against diffusion-based image editing, no directly comparable methods exist under the same setting. Therefore, we establish baselines for universal image immunization by adapting three representative image-specific methods:

Table 2: Black-box immunization transferability on Stable Diffusion V1.4, V2.0 (Rombach et al., 2022), and InstructPix2Pix (Brooks et al., 2023). The surrogate model is Stable Diffusion V1.5.

| Model | Stable Diffusion V1.4 | | | | | | Stable Diffusion V2.0 | | | | | | InstructPix2Pix | | | | | |
|---|---|---|---|---|---|---|---|---|---|---|---|---|---|---|---|---|---|---|
| Method | clean | Enc. | Emb. | Map | Ours$_{DF}$ | Ours | clean | Enc. | Emb. | Map | Ours$_{DF}$ | Ours | clean | Enc. | Emb. | Map | Ours$_{DF}$ | Ours |
| PSNR ↓ | – | 18.08 | 17.00 | 17.51 | 14.66 | **14.23** | – | 14.33 | 14.25 | 13.89 | 13.22 | **12.17** | – | 18.18 | 17.11 | 17.12 | 16.06 | **15.36** |
| SSIM ↓ | – | 0.634 | 0.467 | 0.584 | 0.345 | **0.318** | – | 0.417 | 0.335 | 0.355 | 0.314 | **0.240** | – | 0.571 | 0.521 | 0.498 | 0.474 | **0.418** |
| VIFp ↓ | – | 0.239 | 0.162 | 0.210 | 0.091 | **0.075** | – | 0.122 | 0.099 | 0.107 | 0.085 | **0.057** | – | 0.205 | 0.207 | 0.204 | 0.164 | **0.135** |
| FSIM ↓ | – | 0.787 | 0.732 | 0.759 | 0.656 | **0.641** | – | 0.654 | 0.644 | 0.624 | 0.608 | **0.555** | – | 0.816 | 0.790 | 0.805 | 0.760 | **0.713** |
| LPIPS ↑ | – | 0.348 | 0.492 | 0.399 | 0.565 | **0.605** | – | 0.510 | 0.579 | 0.556 | 0.597 | **0.660** | – | 0.372 | 0.419 | 0.442 | 0.464 | **0.527** |
| Feat. Sim. (C) ↓ | 0.743 | 0.707 | 0.690 | 0.702 | 0.687 | **0.675** | 0.694 | 0.652 | 0.647 | 0.643 | 0.629 | **0.616** | 0.810 | 0.787 | 0.756 | 0.769 | 0.752 | **0.729** |
| Feat. Sim. (D) ↓ | 0.531 | 0.436 | 0.399 | 0.421 | 0.374 | **0.346** | 0.408 | 0.275 | 0.274 | 0.257 | 0.224 | **0.191** | 0.658 | 0.646 | 0.590 | 0.647 | 0.552 | **0.527** |

Table 3: Robustness evaluation against various purification methods.

| Purification | Clean | JPEG compression | | | | Adv.Cleaner | | | | Conditional DiffPure | | | | Noisy Upscaling | | | |
|---|---|---|---|---|---|---|---|---|---|---|---|---|---|---|---|---|---|
| Method | Sim. | Enc. | Emb. | Map | Ours | Enc. | Emb. | Map | Ours | Enc. | Emb. | Map | Ours | Enc. | Emb. | Map | Ours |
| PSNR ↓ | – | 20.96 | 18.67 | 18.21 | 17.64 | 22.38 | 19.56 | 18.51 | **18.03** | 16.64 | 16.54 | 16.40 | **14.99** | 18.22 | 17.99 | 17.90 | **17.20** |
| SSIM ↓ | – | 0.736 | 0.600 | **0.547** | 0.560 | 0.801 | 0.668 | **0.578** | 0.590 | 0.524 | 0.462 | 0.475 | **0.371** | 0.558 | 0.551 | 0.548 | **0.482** |
| VIFp ↓ | – | 0.337 | 0.230 | 0.210 | **0.198** | 0.379 | 0.253 | 0.214 | **0.201** | 0.173 | 0.148 | 0.165 | **0.102** | 0.195 | 0.195 | 0.190 | **0.165** |
| FSIM ↓ | – | 0.848 | 0.783 | 0.771 | **0.757** | 0.886 | 0.817 | 0.788 | **0.772** | 0.733 | 0.717 | 0.718 | **0.664** | 0.762 | 0.760 | 0.757 | **0.732** |
| LPIPS ↑ | – | 0.262 | 0.384 | 0.410 | **0.419** | 0.192 | 0.326 | 0.394 | **0.398** | 0.423 | 0.466 | 0.455 | **0.568** | 0.383 | 0.384 | 0.389 | **0.427** |
| Feat. Sim. (C) ↓ | 0.744 | 0.720 | 0.710 | 0.699 | **0.694** | 0.731 | 0.724 | 0.707 | **0.700** | 0.707 | 0.707 | 0.701 | **0.683** | 0.728 | 0.730 | 0.730 | **0.723** |
| Feat. Sim. (D) ↓ | 0.534 | 0.481 | 0.460 | 0.438 | **0.432** | 0.507 | 0.488 | 0.461 | **0.445** | 0.448 | 0.457 | 0.438 | **0.376** | 0.513 | 0.505 | 0.500 | **0.490** |

(i) the encoder attack from PhotoGuard (Salman et al., 2023) (*Encoder*), (ii) AdvPaint (Jeon et al., 2025), which alters intermediate embeddings of self-attention and cross-attention (*Embedding*), and (iii) Semantic Attack (Lo et al., 2024), which manipulates cross-attention maps (*Map*). Detailed descriptions of the modifications are provided in Appendix 7.4. To further assess the effectiveness and efficiency of our approach, we also compare against state-of-the-art image-specific immunization methods, including PhotoGuard (Salman et al., 2023)—with its Encoder Attack (EA) and Diffusion Attack (DA)—the Semantic Attack (SA) (Lo et al., 2024), and EditShield (ES) (Chen et al., 2024) for image editing.

**Implementation Details.** In our experiments, we train UAPs on the Stable Diffusion V1.5 (Rombach et al., 2022) model available on Hugging Face. Following prior work on classifier UAPs (Lee et al., 2025; Liu et al., 2023b), we set $\epsilon = 10/255$ to limit visible artifacts, as a single perturbation must generalize across inputs. For image-specific methods, we adopt the convention (Salman et al., 2023; Lo et al., 2024) with $\epsilon = 16/255$. The number of optimization iterations is set to 100 for image-specific methods, while ours is trained for 20 epochs. The timestep parameter in Algorithm 1 is set to $T = \{5, 10, 15, 20, 25\}$. We adopt the default setting of Stable Diffusion V1.5 for all remaining hyperparameters. To evaluate black-box transferability, we further test our trained UAPs on Stable Diffusion V1.4, V2.0 (Rombach et al., 2022), and InstructPix2Pix (Brooks et al., 2023).

**Dataset and Evaluation Protocol.** To train UAPs, we randomly sample 10,000 image-prompt pairs from LAION-2B-en (Schuhmann et al., 2022) in the data-dependent setting. For the data-free setting, we follow prior UAP methods (Lee et al., 2025; Mopuri et al., 2018) by using randomly generated jigsaw puzzle images as training data. For evaluation, due to the absence of an official benchmark for image immunization, we adopt the data generation procedure from (Lo et al., 2024). Specifically, we generate 50 images for each of 10 distinct object classes using the diffusion model (Rombach et al., 2022). For each class, we define two editing prompts: one for object edits, the other for non-object (*i.e.*, background). To better assess generalization, we use a broader set of 10 object classes and a total of 500 images, compared to the 3 classes and 150 images used in (Lo et al., 2024). Target images are defined by arbitrary prompts and generated via a diffusion model, as in dataset construction. We use '*tiger*' for evaluation and '*Ronaldo*' for qualitative visualization.

We also follow the evaluation protocols of prior works (Salman et al., 2023; Lo et al., 2024; Chen et al., 2024), employing five perceptual quality metrics: PSNR, SSIM (Wang et al., 2004), VIFp (Sheikh & Bovik, 2006), FSIM (Zhang et al., 2011), and LPIPS (Zhang et al., 2018) and two semantic alignment metrics that measure the feature similarity between source and edited images using CLIP (Radford et al., 2021) and DINO (Zhang et al., 2023a) representations. All results are averaged over 10 random seeds to ensure statistical robustness. Further details on dataset construction, visualizations on various targets, and evaluation protocols are provided in Appendix 7.

Table 4: Comparison with image-specific methods in the white-box setting on Stable Diffusion v1.5. *Ours* denotes the full method, while *Ours$_{DF}$* uses only the target semantic injection loss in a data-free setting. The best result is shown in bold; the second-best is underlined.

| Method | | Image-specific | | | | Universal | |
|---|---|---|---|---|---|---|---|
| | | EA | DA | SA | ES | Ours$_{DF}$ | Ours |
| Metrics | PSNR ↓ | 14.75±0.715 | 14.71±0.951 | 16.28±1.547 | 16.04±0.820 | 14.68±0.035 | **14.19**±0.059 |
| | SSIM ↓ | 0.386±0.059 | 0.382±0.082 | 0.431±0.102 | 0.335±0.067 | 0.378±0.002 | **0.332**±0.003 |
| | VIFp ↓ | 0.088±0.025 | 0.106±0.037 | 0.138±0.052 | 0.106±0.031 | 0.106±0.001 | **0.082**±0.001 |
| | FSIM ↓ | **0.637**±0.023 | 0.660±0.037 | 0.714±0.051 | 0.691±0.028 | 0.666±0.002 | 0.642±0.003 |
| | LPIPS ↑ | 0.584±0.034 | 0.552±0.050 | 0.490±0.064 | 0.553±0.041 | 0.557±0.002 | **0.606**±0.003 |
| | Feat. Sim. (C) ↓ | 0.677±0.002 | **0.662**±0.002 | 0.705±0.002 | 0.684±0.003 | 0.685±0.003 | 0.670±0.002 |
| | Feat. Sim. (D) ↓ | 0.373±0.005 | **0.333**±0.006 | 0.447±0.006 | 0.387±0.004 | 0.376±0.003 | 0.345±0.002 |
| Test-Time | Latency (sec) | 8.01 | 212.46 | 55.66 | 10.48 | ∼0 | ∼0 |
| Cost (per image) | GPU Usage (GB) | 6.51 | 30.84 | 9.08 | 7.02 | 0 | 0 |

## 5.2 MAIN RESULTS

**Comparison with Universal Immunization Baselines.** We first evaluate our method against the universal extensions of existing state-of-the-art image-specific immunization approaches. As shown in Table 1, our method consistently outperforms these baselines by a clear margin. Notably, even our data-free variant achieves competitive results, ranking second-best across most metrics, where it highlights the effectiveness and practicality of our approach.

We further evaluate our UAP's transferability across editing models, including Stable Diffusion V1.4, V2.0 (Rombach et al., 2022) and InstructPix2Pix (Brooks et al., 2023). As shown in Table 2, our method consistently outperforms universal baselines in the black-box setting. Notably, higher LPIPS scores indicate substantial perceptual and semantic shifts, likely caused by our semantic injection, which steers the editing process to follow the injected target rather than the original content. These results across models highlight the strong transferability of our semantic injection approach.

**Robustness Evaluation against Purification.** We evaluate the robustness of our method and the universal baselines against post-processing noise purification techniques. Specifically, we consider two general-purpose defenses, JPEG compression (Guo et al., 2018) and AdverseCleaner (Diaz, 2025), as well as Conditional DiffPure and Noisy Upscaling (Hönig et al., 2025), which are tailored to perturbation purification in diffusion models. Importantly, diffusion-based purification methods such as Conditional DiffPure and Noisy Upscaling act as adaptive defenses (Tramer et al., 2020) against our approach. Specifically, while our method injects target semantics into the original image, these methods attempt to restore the original semantics by reconstructing or upscaling the content using the original image prompt. Thus, they can be viewed as adaptive strategies that assume awareness of our immunization mechanism. As shown in Table 3, our carefully designed semantic injection approach consistently outperforms the universal baselines after purification, demonstrating its robustness against defenses.

**Comparison with Image-specific Immunization Methods.** We further compare our method against existing image-specific immunization methods on image editing tasks in a white-box setting, considering both performance and test-time efficiency. As shown in Table 4, our universal method significantly outperforms image-specific approaches, except FSIM. Notably, even under stricter constraints, such as the absence of real data and a smaller perturbation budget, *Inj$_{DF}$* achieves competitive performance compared to existing image-specific methods. Unlike prior methods that require costly per-instance GPU optimization, our method generates immunized images in near-zero time via a simple element-wise addition, ensuring both efficiency and practicality.

**Qualitative Results.** We first present qualitative results on image editing tasks, comparing our method with universal baselines (*i.e.* Encoder, Embedding, and Map). As shown in Figure 3, the edited outputs clearly ignore the original source content. For instance, prompts applied to the *deer* and *bus* images yield unrelated human figures. In the case of *cow* and *man*, the results remain related to their corresponding prompts, but differ significantly from the originals and instead resemble the injected target semantics. This effect is further supported by the similarity between outputs from immunized images and those edited using the UAP alone (Only UAP), as well as the attention maps in Figure 2(c), highlighting the dominant influence of the injected target. More qualitative results are provided in Appendix 10.

Table 5: Ablation study for our proposed method. *Inj$_{DF}$*, *Inj*, and *Inj+Sup* denote results obtained using UAPs trained with $\mathcal{L}_{\text{inj}}$ in the data-free setting, data-dependent setting, and the combination of $\mathcal{L}_{\text{inj}}$ and $\mathcal{L}_{\text{sup}}$, respectively. * indicates the white-box result.

| Model | Stable Diffusion V1.4 | | | | Stable Diffusion V1.5* | | | | Stable Diffusion V2.0 | | | | InstructPix2Pix | | | |
|---|---|---|---|---|---|---|---|---|---|---|---|---|---|---|---|---|
| Method | Clean | Inj$_{DF}$ | Inj | Inj+Sup | Clean | Inj$_{DF}$ | Inj | Inj+Sup | Clean | Inj$_{DF}$ | Inj | Inj+Sup | Clean | Inj$_{DF}$ | Inj | Inj+Sup |
| PSNR ↓ | – | 14.66 | 14.33 | 14.23 | – | 14.68 | 14.41 | 14.19 | – | 12.33 | 12.13 | 12.04 | – | 16.06 | 15.61 | 15.36 |
| SSIM ↓ | – | 0.345 | 0.332 | 0.318 | – | 0.378 | 0.367 | 0.332 | – | 0.253 | 0.245 | 0.235 | – | 0.474 | 0.433 | 0.418 |
| VIFp ↓ | – | 0.091 | 0.080 | 0.075 | – | 0.106 | 0.096 | 0.082 | – | 0.058 | 0.053 | 0.051 | – | 0.164 | 0.141 | 0.135 |
| FSIM ↓ | – | 0.656 | 0.646 | 0.641 | – | 0.666 | 0.655 | 0.642 | – | 0.573 | 0.567 | 0.564 | – | 0.760 | 0.719 | 0.713 |
| LPIPS ↑ | – | 0.565 | 0.591 | 0.605 | – | 0.557 | 0.585 | 0.606 | – | 0.643 | 0.658 | 0.665 | – | 0.464 | 0.514 | 0.527 |
| Feat. Sim. (C) ↓ | 0.743 | 0.687 | 0.681 | 0.675 | 0.744 | 0.685 | 0.680 | 0.673 | 0.694 | 0.629 | 0.620 | 0.616 | 0.810 | 0.752 | 0.731 | 0.729 |
| Feat. Sim. (D) ↓ | 0.531 | 0.374 | 0.367 | 0.346 | 0.534 | 0.376 | 0.372 | 0.345 | 0.408 | 0.224 | 0.204 | 0.191 | 0.658 | 0.552 | 0.536 | 0.527 |

Table 6: Ablation study on target selection. * denotes the prompt used in our main experiments.

| Target | Clean | Ronaldo | Tiger* | Sunflower | Peacock | Mandala | Avg. | Std. |
|---|---|---|---|---|---|---|---|---|
| PSNR ↓ | – | $14.69_{\pm 0.033}$ | $14.19_{\pm 0.058}$ | $14.32_{\pm 0.044}$ | $14.22_{\pm 0.055}$ | $14.25_{\pm 0.033}$ | 14.33 | 0.199 |
| SSIM ↓ | – | $0.383_{\pm 0.002}$ | $0.332_{\pm 0.003}$ | $0.340_{\pm 0.002}$ | $0.308_{\pm 0.003}$ | $0.301_{\pm 0.003}$ | 0.332 | 0.029 |
| VIFp ↓ | – | $0.095_{\pm 0.001}$ | $0.082_{\pm 0.001}$ | $0.088_{\pm 0.001}$ | $0.092_{\pm 0.001}$ | $0.078_{\pm 0.001}$ | 0.087 | 0.006 |
| FSIM ↓ | – | $0.660_{\pm 0.001}$ | $0.642_{\pm 0.002}$ | $0.651_{\pm 0.002}$ | $0.639_{\pm 0.001}$ | $0.641_{\pm 0.001}$ | 0.646 | 0.008 |
| LPIPS ↑ | – | $0.576_{\pm 0.002}$ | $0.606_{\pm 0.003}$ | $0.610_{\pm 0.003}$ | $0.598_{\pm 0.003}$ | $0.621_{\pm 0.003}$ | 0.602 | 0.015 |
| Feat. Sim. (C) ↓ | 0.744 | $0.677_{\pm 0.003}$ | $0.670_{\pm 0.002}$ | $0.679_{\pm 0.001}$ | $0.681_{\pm 0.002}$ | $0.673_{\pm 0.002}$ | 0.676 | 0.004 |
| Feat. Sim. (D) ↓ | 0.534 | $0.357_{\pm 0.002}$ | $0.345_{\pm 0.004}$ | $0.344_{\pm 0.001}$ | $0.346_{\pm 0.003}$ | $0.347_{\pm 0.001}$ | 0.347 | 0.004 |

**Limitations and Discussions.** A potential concern is reverse-engineering of protections. However, our method generates diverse UAPs even for the same prompt, producing varied protection patterns that hinder reverse-engineering, as supported by Table 6 and Appendix 11. Further analysis of limitations, discussions and societal impacts is provided in Appendix 11 and 12.

## 5.3 ABLATION STUDIES

**Impact of Each Proposed Components.** In Table 5, we report immunization performance under three settings to show the impact of real data and our proposed losses: *Inj$_{DF}$* applies the target semantic injection loss in a data-free setting; *Inj* uses the same loss with real data; *Inj+Supp* adds the source semantic suppression loss. The improvement from *Inj$_{DF}$* to *Inj* further shows that real data significantly improves the model's ability to internalize the adversarial objective. Adding the suppression loss further improves robustness and transferability by mitigating residual source semantics. These results highlight the complementary benefits of target semantic injection, data realism, and semantic disentanglement in effective black-box immunization. For all ablation studies, we adopt Stable Diffusion v1.5 as the surrogate model for UAP generation.

**Diverse Target Analysis.** To demonstrate the target-independency of our method, we arbitrarily select five diverse targets, *e.g.*, *'Ronaldo', 'Tiger', 'Sunflower', 'Peacock'*, and *'Mandala'*, and evaluate performance across them. As shown in Table 6, our method consistently achieves strong defense results with low variance across these targets, indicating its robustness to the choice of target. This stability, along with high mean performance, suggests that our UAP generalizes well regardless of the target semantics. We include visualizations of the generated UAPs and qualitative editing results for each target in the Appendix.

## 6 CONCLUSION

We introduced the first universal image immunization framework that defends against diffusion-based editing by injecting target semantics through a targeted UAP. By incorporating two optimization objectives–target semantic injection and source semantic suppression–the method effectively overrides the source content while suppressing the original semantics, guiding edits toward the injected target. As it requires only a simple additive operation at test time, the method enables highly efficient and model-agnostic deployment. Our universal immunization approach significantly outperforms all universal baselines adapted from prior image-specific methods, remains effective in data-free settings, and demonstrates strong competitiveness against image-specific approaches, underscoring its practical applicability.

## REPRODUCIBILITY STATEMENT

For reproducibility, we have included the source code in the supplementary materials and have provided pseudocode for UAP training and test-time immunization in Algorithms 1 and 2, respectively. Detailed dataset settings and hyperparameters are reported in Appendix 7, and pseudocode for the data-free setting is provided in Algorithm 3 of Appendix 7.3. Our code, evaluation datasets, and seeds will be publicly released if the paper is accepted.

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

# Appendix

## Universal Image Immunization against Diffusion-based Image Editing via Semantic Injection

## 7    EXPERIMENTAL DETAILS

### 7.1    DATASETS AND EVALUATION PROTOCOLS

**Evaluation Dataset for Image Editing** ($D_E$).    As mentioned in Sec. 5 of the main manuscript, we constructed evaluation datasets for image editing, by following the setup of prior work (Salman et al., 2023; Lo et al., 2024). Specifically, we generated 50 samples for each of 10 distinct object classes using Stable Diffusion V3 (Esser et al., 2024). Compared to the previous setup (Lo et al., 2024), which used only 3 object classes (50×3 classes = 150 images), our expanded dataset allows for a more comprehensive evaluation of the generalization capability of the universal adversarial perturbation (UAP) across a broader range of visual concepts. Generated samples of each class are visualized in Figure 4. Furthermore, we designed two editing prompts for each class, each describing the manipulation of either a specific object or the background, as shown in Table 7. For convenience, we refer to this entire image-text pair dataset (50×10 concepts = 500 images) as $D_E$.

| Bus | Cat | Cow | Deer | Dog | Horse | Man | Truck | Woman | Zebra |

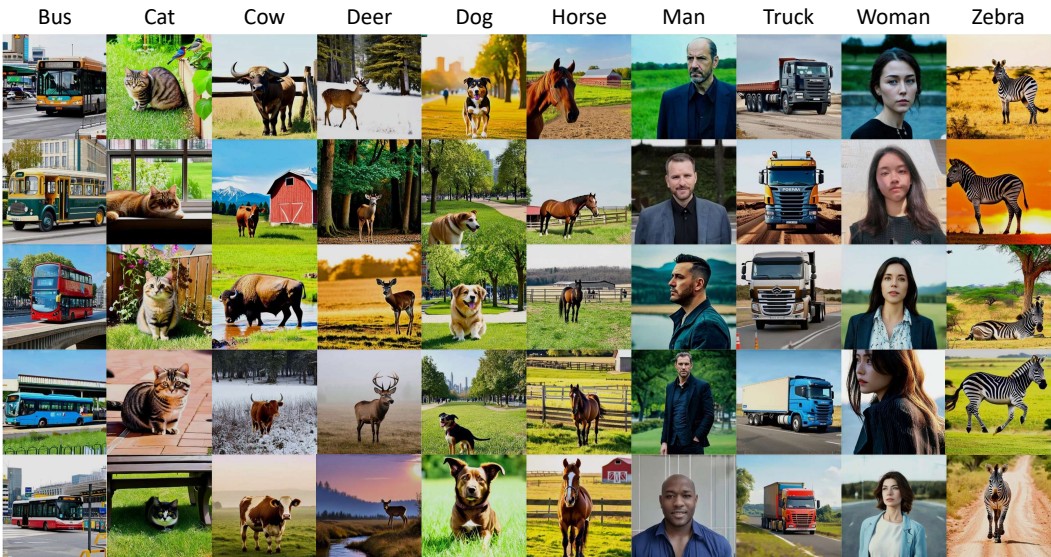

Figure 4: Examples of our generated evaluation dataset.

Table 7: Selected object classes and prompts for the evaluation dataset. Each object class is associated with two editing prompts.

| Object Class | Prompt 1 | Prompt 2 |
|---|---|---|
| bus | *"A photo of a truck"* | *"A bus in the desert"* |
| truck | *"A photo of a bus"* | *"A truck in the park"* |
| cat | *"A photo of a dog"* | *"A cat on the sand"* |
| dog | *"A photo of a cat"* | *"A dog with a ball"* |
| zebra | *"A photo of a deer"* | *"A zebra with a horse"* |
| deer | *"A photo of a zebra"* | *"A deer in the city"* |
| man | *"A photo of a woman"* | *"A man with a hat"* |
| woman | *"A photo of a man"* | *"A woman in the beach"* |
| cow | *"A photo of a horse"* | *"A cow is on mud"* |
| horse | *"A photo of a cow"* | *"A horse and a sheep"* |

**Evaluation Dataset for Image Inpainting** ($D_I$).    To verify whether our method extends beyond immunization for image editing to image inpainting, we compare it with two image inpainting methods: DiffusionGuard (DG) (Choi et al., 2025) and AdvPaint (AP) (Jeon et al., 2025). For evaluation, we construct the inpainting dataset $D_I$ by selecting 3 images from each of the 10 object classes in $D_E$ and generating a binary mask for each image following the DG's inpainting protocol. The white regions in the binary masks are then filled by the inpainting models.

**Evaluation Protocols.** We compare ours with other methods via five widely used perceptual image quality metrics: PSNR, SSIM (Wang et al., 2004), VIFp (Sheikh & Bovik, 2006), FSIM (Zhang et al., 2011), and LPIPS (Zhang et al., 2018) as described in the manuscript. Specifically, PSNR evaluates pixel-level fidelity based on mean squared error. SSIM compares structural information such as luminance, contrast, and texture. VIFp measures the amount of visual information preserved, incorporating models of natural image statistics and human perception. FSIM assesses feature-level similarity using phase congruency and gradient magnitude, which are believed to align well with human visual sensitivity. LPIPS compares deep feature representations from neural networks, offering a perceptual similarity measure that correlates closely with human judgments. These metrics are computed between editing results of immunized and non-immunized images.

We follow the default settings of each pipeline for evaluation. For the image-to-image pipeline of Stable Diffusion V1.5, we set the number of inference steps to 50, the strength to 0.8, and the guidance scale to 7.5. For Stable Diffusion V1.5 inpainting, we use an inference step of 50, a guidance scale of 7.5, and a strength of 1.0. To account for the stochastic nature of diffusion models, we randomly select 10 seeds and report the average of the quantitative results.

**Target Generation** To generate target images, we use Stable Diffusion (Rombach et al., 2022) with the prompt "a photo of *<target>*." As target concepts, we generate five representative examples—*Ronaldo*, *Tiger*, *Sunflower*, *Peacock*, and *Mandala*—and evaluate our method on them. However, our approach is not limited to this particular set and can, in principle, be applied to any target concept, where multiple target images may also be considered for the same concept. Figure 5 shows examples of target images and their corresponding UAPs for five randomly selected target objects. Figure 6 further visualizes UAPs trained with different target images generated with the same target prompt, 'tiger.'

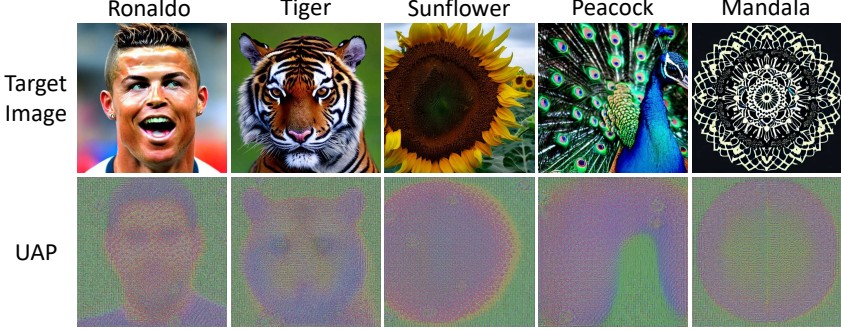

Figure 5: Visualization for targets and their corresponding UAPs. UAPs are scaled for visualization.

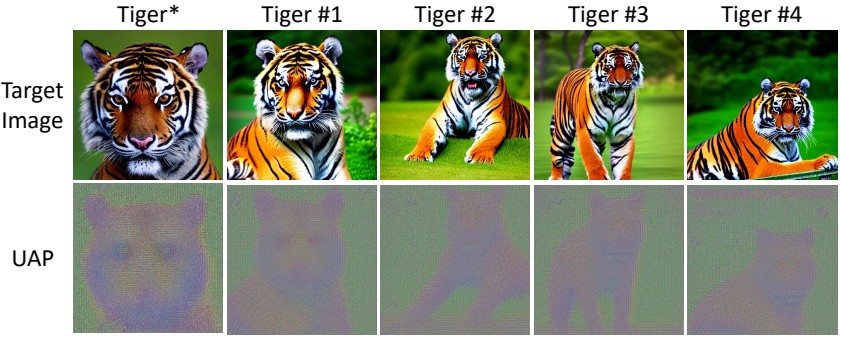

Figure 6: Visualization for diverse target images on the same prompt ('tiger') and their corresponding UAPs. * indicates the target image used in our experiments. UAPs are scaled for visualization.

## 7.2 TRAINING SETUP

For training our UAP, we set the attack stepsize $s = 1/255$, constraint parameter $\epsilon = 10/255$, timestep set $K = \{5, 10, 15, 20, 25\}$, inference steps to 50, and the guidance scale in diffusion models to 7.5. For the baselines, we adopt the same diffusion process settings, including 50 inference

steps and a guidance scale of 7.5. The number of attack iterations is set to 100, and all other configurations follow those specified in the original implementations.

### 7.3 DATA-FREE UNIVERSAL IMMUNIZATION

In this subsection, we describe how to train a targeted universal adversarial perturbation (UAP) in a fully data-free manner against diffusion-based editing models. Following the strategy of TRM-UAP (Liu et al., 2023b), we generate random jigsaw puzzles and apply a mean filter to smooth the edges between regions, resembling real-world training samples. We also adopt a curriculum learning scheme that starts with simple artificial images and gradually introduces more complex ones, controlled by a distribution parameter and the training iteration. Figure 7 demonstrates examples of random jigsaw puzzles, where the complexity of the artificial images increases from left to right. The overall training procedure for our data-free targeted UAP is presented in Algorithm 3.

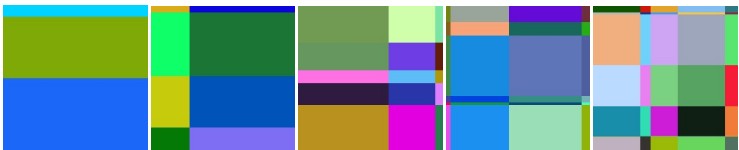

Figure 7: Examples of jigsaw puzzle images.

---

**Algorithm 3** Data-free Targeted UAP Training

---

**Input:** Random data prior $\mathcal{D}_r$, target image $x_{tar}$, target prompt token embedding $t_{tar}$, surrogate diffusion model $\mathcal{M}$, diffusion timestep set $K = \{k_1, k_2, ..., k_n\}$, training epoch $N$, attack stepsize $s$, constraint parameter $\epsilon$

**Output:** Universal adversarial perturbation $\delta$.

1: Initialize $\delta \sim \mathcal{U}(-\epsilon, \epsilon)$
2: **while** $i < N$ **do**
3:     $i = i + 1$
4:     **for** each training sample **do**
5:         Sample $x_r \sim \mathcal{D}_r$
6:         $grad\_total \leftarrow 0$
7:         $x_{adv} \leftarrow clip(x_r + \delta, 0, 1)$
8:         **for** $k$ in $K$ **do**
9:             // Inject forward noise to both target and immunized images
            $x_{tar}^k \leftarrow x_{tar} + noise\_scheduler(k)$
            $x_{adv}^k \leftarrow x_{adv} + noise\_scheduler(k)$
10:          // Compute target semantic injection loss using Eq. (4)
            $\mathcal{L} \leftarrow \mathcal{L}_{inj}$
11:         $grad\_total \leftarrow grad\_total + \nabla\mathcal{L}$
12:         **end for**
13:         // Update and clip perturbation $\delta$
        $\delta \leftarrow \delta - s \cdot sign(grad\_total)$
        $\delta \leftarrow \min(\epsilon, \max(\delta, -\epsilon))$
14:     **end for**
15: **end while**
16: **return** $\delta$

---

### 7.4 IMPLEMENTATION DETAILS FOR UNIVERSAL BASELINES

We construct universal baselines by adapting existing image-specific methods to the UAP setting with their original loss functions, ensuring a fair comparison, as we are the first to propose a universal immunization framework against diffusion-based image editing. Specifically, we adopt three loss functions from prior image-specific studies: Encoder Attack (Salman et al., 2023), AdvPaint (Jeon et al., 2025), and Semantic Attack (Lo et al., 2024), as follows:

- **Encoder Attack (Encoder)**: following the original Encoder Attack's strategy, we minimize the $L_2$ distance between the latent representation of an uninformative black reference image and that of the perturbed training image by UAP.

- **Embedding Attack (Embedding)**: we maximize an $L_2$ loss between the intermediate representations (*i.e.* query embedding in cross-attention layers and query/key/value embeddings

in self-attention layers) of the perturbed image and those of the original, encouraging them to deviate.

- **Attention Map Attack (Map)**: following Semantic Attack (Lo et al., 2024), we minimize the alignment between attention maps and the original image semantics, thereby disrupting semantic consistency.

All three baselines are trained on the same dataset as ours, with identical hyperparameters, including 20 training epochs, step size $s = 1/255$, and perturbation budget $\epsilon = 10/255$.

## 8 ADDITIONAL EXPERIMENTAL RESULTS

### 8.1 BLACK-BOX EVALUATION WITH IMAGE-SPECIFIC IMMUNIZATION

To demonstrate the superiority of our method, we evaluate its black-box transferability against existing state-of-the-art image-specific immunization methods (Salman et al., 2023; Lo et al., 2024). Notably, our method achieves competitive or even superior performance to the image-specific methods, even under a small perturbation budget ($\epsilon = 16/255$ vs. $\epsilon = 10/255$) as shown in Table 8. These results further confirm that our universal immunization not only eliminates the need for per-image optimization with GPU (see in Table 4), but also provides a practical and efficient image safeguarding.

Table 8: Black-box immunization transferability on Stable Diffusion V1.4, V2.0 Rombach et al. (2022), and InstructPix2Pix Brooks et al. (2023). The surrogate model for each method is Stable Diffusion V1.5.

| Model | | Stable Diffusion V1.4 | | | | | | Stable Diffusion V2.0 | | | | | | InstructPix2Pix | | | | | |
|---|---|---|---|---|---|---|---|---|---|---|---|---|---|---|---|---|---|---|---|---|
| Method | Clean | *Image-specific* | | | | *Universal* | | Clean | *Image-specific* | | | | *Universal* | | Clean | *Image-specific* | | | | *Universal* | |
| | | EA | DA | SA | ES | Ours$_{DF}$ | Ours | | EA | DA | SA | ES | Ours$_{DF}$ | Ours | | EA | DA | SA | ES | Ours$_{DF}$ | Ours |
| PSNR ↓ | – | 15.46 | 15.54 | 14.91 | 16.24 | 15.60 | **14.18** | – | 12.48 | 12.73 | 12.79 | 13.74 | 13.22 | **12.17** | – | 16.02 | 15.89 | 17.26 | **14.77** | 16.06 | 15.36 |
| SSIM ↓ | – | 0.444 | 0.457 | 0.336 | 0.408 | 0.467 | **0.332** | – | 0.298 | 0.304 | 0.271 | 0.317 | 0.314 | **0.240** | – | 0.445 | 0.409 | 0.499 | **0.396** | 0.474 | 0.418 |
| VIFp ↓ | – | 0.119 | 0.143 | 0.088 | 0.141 | 0.146 | **0.083** | – | 0.074 | 0.085 | 0.060 | 0.092 | 0.085 | **0.057** | – | 0.133 | 0.134 | 0.189 | **0.128** | 0.164 | 0.135 |
| FSIM ↓ | – | 0.664 | 0.695 | 0.666 | 0.691 | 0.702 | **0.642** | – | 0.558 | 0.586 | 0.632 | 0.608 | 0.608 | **0.555** | – | 0.747 | 0.755 | 0.795 | **0.702** | 0.760 | 0.713 |
| LPIPS ↑ | – | 0.545 | 0.507 | 0.549 | 0.510 | 0.509 | **0.604** | – | 0.634 | 0.611 | 0.626 | 0.593 | 0.597 | **0.660** | – | 0.509 | **0.533** | 0.433 | 0.529 | 0.464 | 0.527 |
| Feat. Sim. (C) ↓ | 0.743 | **0.663** | 0.675 | 0.701 | 0.704 | 0.687 | 0.675 | 0.694 | **0.606** | 0.628 | 0.656 | 0.651 | 0.629 | 0.616 | 0.810 | 0.774 | 0.788 | 0.773 | **0.721** | 0.752 | 0.729 |
| Feat. Sim. (D) ↓ | 0.531 | **0.335** | 0.370 | 0.445 | 0.426 | 0.374 | 0.346 | 0.408 | **0.188** | 0.210 | 0.307 | 0.303 | 0.224 | 0.191 | 0.658 | 0.655 | 0.643 | 0.632 | **0.522** | 0.552 | 0.527 |

### 8.2 BLACK-BOX EVALUATION ON DiT-BASED DIFFUSION MODEL

To evaluate the robustness of our method across different model architectures, we perform experiments using diffusion transformer (DiT) model. Specifically, we adopt Stable Diffusion V3 (Esser et al., 2024), configured to generate images at a resolution of $512 \times 512$, consistent with our experimental setup, for the image editing task. All universal and image-specific perturbations are generated on Stable Diffusion V1.5 (Rombach et al., 2022).

In Table 9, while all methods, including ours, tend to perform worse on transformer-based diffusion models compared to U-Net-based ones, our method achieves superior performance for image immunization compared to other methods. Specifically, compared with the universal baselines, our method achieves significantly stronger immunization performance across all metrics, clearly demonstrating the effectiveness of our approach. Even when compared with image-specific methods such as EA, DA (Salman et al., 2023), and SA (Lo et al., 2024), our method attains the highest performance in every metric except SSIM, highlighting its robustness beyond the universal setting. These results confirm that our method not only provides superior immunization but also exhibits excellent cross-architecture transferability.

Table 9: Black-box immunization performance on Stable Diffusion V3 Esser et al. (2024) for image editing using $D_E$. All perturbations are generated using Stable Diffusion V1.5.

| Method | Image-specific | | | Universal | | | |
|---|---|---|---|---|---|---|---|
| metric | EA | DA | SA | Enc. | Que. | Map | Ours |
| PSNR ↓ | 19.24 | 18.71 | 20.01 | 19.96 | 19.39 | 18.92 | **17.63** |
| SSIM ↓ | 0.653 | **0.620** | 0.715 | 0.695 | 0.653 | 0.636 | 0.624 |
| VIFp ↓ | 0.234 | 0.230 | 0.301 | 0.284 | 0.264 | 0.246 | **0.227** |
| FSIM ↓ | 0.783 | 0.772 | 0.806 | 0.804 | 0.786 | 0.777 | **0.756** |
| LPIPS ↑ | 0.452 | 0.465 | 0.384 | 0.405 | 0.428 | 0.448 | **0.467** |

## 8.3 QUANTITATIVE COMPARISON WITH IMAGE INPAINTING IMMUNIZATION METHODS

We evaluate our method on the image inpainting task using the inpainting dataset $D_I$, and compare its performance with two baselines: DiffusionGuard (DG) (Choi et al., 2025) and AdvPaint (AP) (Jeon et al., 2025), specifically designed for image inpainting immunization, under the same perturbation budget $\epsilon = 16/255$. Target models are U-Net based Stable Diffusion inpainting V1.5 and V2.0, and DiT-based FLUX (Labs, 2024). Following the inpainting protocol of DiffusionGuard (Choi et al., 2025), all adversarial perturbations are applied to the designated sensitive region using the binary mask.

As shown in Tables 10, our method achieves competitive performance against dedicated inpainting immunization approaches in both white-box and U-Net-based black-box (SD V2) settings, demonstrating strong generalizability across diffusion-based manipulation tasks—even though it was not specifically designed for inpainting. Notably, our method achieves more robust results than prior approaches on the DiT-based image inpainting pipeline. We believe this is due to the stronger semantic overriding effect induced by our targeted UAP, which suppresses the source content more effectively than previous untargeted image-specific methods. Furthermore, Table 11 further demonstrates that our method achieves this performance with negligible test-time cost, in contrast to other approaches, making it practical for deployment.

Table 10: Image immunization performance on Stable Diffusion V1.5, V2.0, and FLUX for image inpainting, evaluated on $D_I$. * denotes white-box surrogate model and † indicates methods specifically designed for image-specific inpainting immunization, which requires masks to generate perturbations, whereas *Ours* uses a single perturbation across all images without mask-based training.

| Model | Stable Diffusion 1.5* | | | Stable Diffusion 2.0 | | | FLUX | | |
|---|---|---|---|---|---|---|---|---|---|
| Metric | DG† | AP† | Ours | DG† | AP† | Ours | DG† | AP† | Ours |
| PSNR ↓ | 18.10 | **15.16** | 17.51 | 18.48 | **15.27** | 17.24 | 25.84 | 22.67 | **20.60** |
| SSIM ↓ | 0.615 | **0.499** | 0.528 | 0.643 | **0.519** | 0.534 | 0.810 | 0.691 | **0.598** |
| VIFp ↓ | 0.309 | **0.238** | 0.239 | 0.325 | 0.246 | **0.239** | 0.446 | 0.344 | **0.258** |
| FSIM ↓ | 0.816 | **0.761** | 0.805 | 0.825 | **0.762** | 0.791 | 0.923 | 0.883 | **0.859** |
| LPIPS ↑ | 0.448 | **0.556** | 0.488 | 0.431 | **0.553** | 0.498 | 0.317 | 0.447 | **0.489** |

Table 11: Comparison of test-time computational cost between image-specific immunization methods and our universal approach. † indicates methods specifically designed for image inpainting immunization, requiring masks to generate perturbations.

| Method | Image-specific | | | | | Universal |
|---|---|---|---|---|---|---|
| | EA | DA | SA | DG† | AP† | Ours |
| Latency (sec) | 8.01 | 212.46 | 55.66 | 156.97 | 183.27 | ∼0 |
| GPU Usage (GB) | 6.51 | 30.84 | 9.08 | 5.13 | 19.59 | 0 |
| Requires mask | ✗ | ✗ | ✗ | ✓ | ✓ | ✗ |

## 8.4 ANALYSIS FOR IMPERCEPTIBILITY

Universal adversarial perturbations generate a single perturbation to deceive multiple images, which makes them more susceptible to visible artifacts compared to image-specific approaches that leverage individual target images. To examine the trade-off between immunization performance and imperceptibility, we further evaluate different perturbation budgets ($\epsilon = 8, 10, 16$), measuring imperceptibility with DISTS (Ding et al., 2020), PSNR, and AHIQ (Lao et al., 2022), alongside standard immunization evaluation metrics. As shown in Table 12, our method achieves a comparable level of imperceptibility to image-specific baselines at smaller budgets (e.g., $\epsilon = 8/255$ and $\epsilon = 10/255$), while at the same time delivering superior immunization performance. These results highlight that our approach effectively balances perceptual quality and robustness without incurring additional test-time costs.

## 8.5 HUMAN EVALUATION

To practically and comprehensively evaluate the utility of our method, we conducted a user study comparing it with image-specific approaches. Since we are the first to propose a universal immu-

Table 12: Trade-off between immunization performance and imperceptibility.

| Category | Metric | EA | DA | SA | Ours$_8$ | Ours$_{10}$ | Ours$_{16}$ |
|---|---|---|---|---|---|---|---|
| Performance | PSNR ↓ | 14.75 | 14.71 | 16.28 | 14.58 | 14.19 | **13.31** |
| | SSIM ↓ | 0.386 | 0.382 | 0.431 | 0.363 | 0.332 | **0.277** |
| | VIFp ↓ | 0.088 | 0.106 | 0.138 | 0.095 | 0.082 | **0.061** |
| | FSIM ↓ | 0.637 | 0.660 | 0.714 | 0.656 | 0.637 | **0.612** |
| | LPIPS ↑ | 0.584 | 0.552 | 0.490 | 0.574 | 0.606 | **0.677** |
| Imperceptibility | DISTS ↓ | 0.311 | 0.243 | 0.164 | **0.157** | 0.181 | 0.246 |
| | PSNR ↑ | 28.67 | 27.78 | 30.82 | **31.37** | 29.41 | 25.29 |
| | AHIQ ↑ | 0.520 | 0.504 | 0.504 | **0.552** | 0.509 | 0.394 |

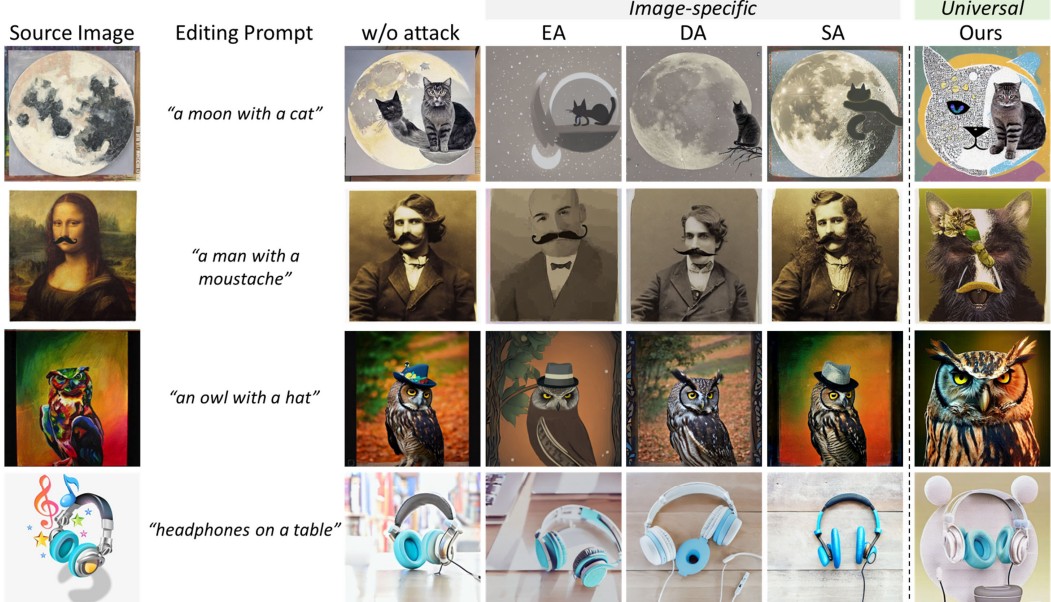

Figure 8: Editing examples of DomainNet images used in the user study after applying immunization across different methods.

nization strategy, these serve as the appropriate baselines. The evaluation was proformed across three dimensions: imperceptibility, effectiveness, and practicality. The study uses 20 representative images: 10 sampled from DomainNet (Peng et al., 2019) (painting and cartoon domains, representing out-of-distribution cases) and 10 from MS-COCO (Lin et al., 2014) (complex, multi-object natural scenes). Each image is perturbed using five methods: Encoder Attack (EA), Diffusion Attack (DA), Semantic Attack (SA), and our approach with perturbation budgets of $\epsilon = 10/255$ and $\epsilon = 16/255$. A total of 20 participants were recruited via Amazon Mechanical Turk (MTurk), with each participant evaluating all perturbed samples in randomized order. For imperceptibility, participants were asked to identify the samples that appeared most visually conspicuous or artifact-heavy, where a higher score indicates that perturbations were more noticeable and thus less imperceptible. For effectiveness, participants ranked the perturbed outputs according to how well they disrupted the intended semantic edit, using a scale from 1 to 5, where higher ranks correspond to stronger immunization. Finally, for practicality, participants rated their willingness to adopt each method in real-world security or safety contexts on a 5-point Likert scale ranging from 1 (not likely) to 5 (very likely), reflecting their subjective judgment of deployability. The human evaluation result in Table 13 reflects a trade-off between invisibility and effectiveness. Interestingly, a similar trend is also observed in the quantitative measurements reported in Table 12. Notably, at $\epsilon = 10/255$, our method strikes a strong balance, demonstrating both high immunization effectiveness and improved imperceptibility as confirmed by human evaluation compared to image-specific baselines. It also receives favorable practicality scores in Table 14, suggesting strong potential for real-world applicability. We additionally provide a visualization of the examples used in the human evaluation in Figure 8 and 9.

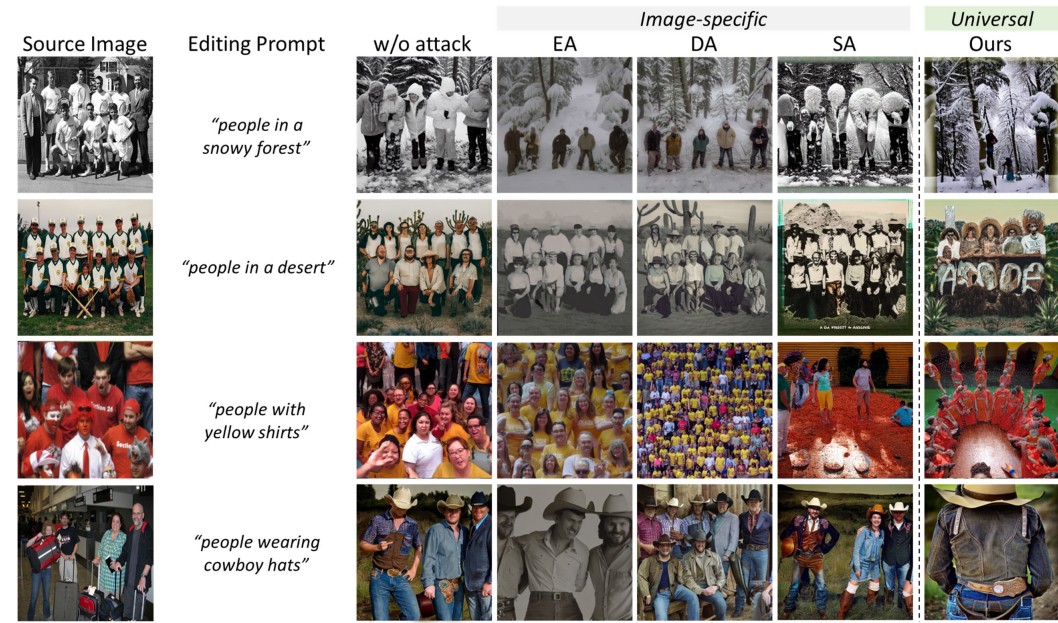

Figure 9: Editing examples of MS-COCO images used in the user study after applying immunization across different methods.

Table 13: User study results on imperceptibility and effectiveness. Imperceptibility is measured as the percentage of votes (lower is better), and effectiveness is measured as the mean ranking score (higher is better).

| Method | EA | DA | SA | Ours$_{10}$ | Ours$_{16}$ |
|---|---|---|---|---|---|
| Imperceptibility (% voting ↓) | 46.75 | 44.25 | **27.00** | 38.00 | 47.00 |
| Effectiveness (mean score ↑) | 2.49±1.33 | 2.23±1.02 | 2.25±1.12 | 3.86 ±0.98 | **4.20**±1.13 |

## 8.6 ROBUSTNESS ON DIVERSE DOMAINS

We evaluate the generality of our method across diverse image types using MS-COCO and DomainNet. MS-COCO contains complex multi-object scenes, while DomainNet serves as an out-of-distribution dataset since it was not seen during UAP training. In Table 15, our method achieves superior performance even when compared to immunization processes that directly leverage source images with a larger perturbation budget of $\epsilon = 16/255$ than ours ($\epsilon = 10/255$). This highlights the robustness of our approach across diverse data domains.

## 9 ADDITIONAL ABLATION STUDY

### 9.1 IMPACT OF TIMESTEP SET.

In our experiments, we set the timestep set to $K = \{5, 10, 15, 20, 25\}$. To analyze the impact of the selected timesteps, we further conduct ablation studies using two alternative timestep sets: $K_1 = \{2, 4, 6, 8, 10\}$ and $K_2 = \{30, 35, 40, 45, 50\}$. Table 16 demonstrates that our method maintains consistently strong performance across different choices of timestep sets, suggesting that it is relatively invariant to the selection of diffusion timesteps.

### 9.2 IMPACT OF DIVERSE CONSTRAINT PARAMETER

To investigate the effect of the $\ell_\infty$-norm constraint parameter $\epsilon$ used in adversarial perturbations on immunization performance, we conduct experiments with $\epsilon = 8/255$ and $16/255$. As shown in Table 12, even with a tighter budget of $\epsilon = 8/255$, our method achieves competitive or superior performance compared to image-specific baselines that use a larger budget of $\epsilon = 16/255$, along

Table 14: User study results on practicality. Participants rated their willingness to adopt our method in security contexts on a 5-point Likert scale.

| Score | 1 | 2 | 3 | 4 | 5 |
|---|---|---|---|---|---|
| Practicality (% voting) | 9 | 27 | 10 | 25 | 29 |

Table 15: Quantitative results on MS-COCO and DomainNet. The best result is shown in bold; the second-best is underlined.

| Dataset | MS-COCO | | | | DomainNet | | | |
|---|---|---|---|---|---|---|---|---|
| Metric | EA | DA | SA | Ours | EA | DA | SA | Ours |
| PSNR ↓ | 14.98 | 14.59 | 16.10 | **14.16** | 14.63 | **14.24** | 16.66 | 14.55 |
| SSIM ↓ | 0.352 | 0.327 | 0.366 | **0.267** | 0.443 | 0.419 | 0.444 | **0.361** |
| VIFp ↓ | 0.066 | 0.078 | 0.115 | **0.060** | **0.079** | 0.111 | 0.130 | 0.083 |
| FSIM ↓ | **0.615** | 0.659 | 0.722 | 0.648 | **0.626** | 0.650 | 0.720 | 0.628 |
| LPIPS ↑ | **0.676** | 0.603 | 0.558 | 0.650 | 0.618 | 0.596 | 0.539 | **0.638** |

with higher imperceptibility, thereby striking a favorable balance between immunization performance and visual quality. We also visualize the editing results in Figure 10, where the UAP is trained on the target '*tiger*'. When the perturbation budget is larger, the editing outputs more clearly deviate from the source image, producing results that are less dependent on the original content. Notably, even at $\epsilon = 8/255$, our method already produces sufficiently source-agnostic results, indicating the strong effectiveness of our approach under smaller perturbation budgets.

### 9.3 ABLATION STUDIES FOR DIVERSE TARGETS

We conduct ablation studies across diverse targets using three variants of the proposed immunization approach: target injection in data-free setting ($Inj_{DF}$), target semantic injection with real data ($Inj$), and injection combined with source semantic suppression ($Inj+Sup$), across various target contents. As shown in Table 17a, 17b, 17c, and 17d, the results consistently show that $Inj$ outperforms $Inj_{DF}$, and that $Inj+Sup$ further improves performance over $Inj$. Importantly, despite relying on pre-computed UAPs with zero test-time cost, our method remains competitive with image-specific baselines. This demonstrates that our *semantic injection loss*, Eq. (4), and *source semantic suppression loss*, Eq. (5), are not only effective across specific examples but also robust and generalizable to arbitrary and diverse target object classes. Furthermore, $Inj+Sup$ consistently outperforms the universal baselines in Table 1, and the comparable or even superior performance of the data-free variant $Inj_{DF}$ further highlights both the effectiveness of semantic injection and its practical advantage of not requiring real-world data.

## 10 MORE QUALITATIVE RESULTS

### 10.1 IMPACT OF DIVERSE TARGETS

We visualize the variation in editing outcomes across different targets in Figure 11. Interestingly, for images protected with semantically distinctive targets such as '*Ronaldo*', '*Tiger*', '*Sunflower*', and '*Peacock*', the editing results tend to retain certain characteristics of the target even after editing. For example, the target '*Ronaldo*' often results in edited outputs that resemble human figures; '*Tiger*' leads to an object whose shape and size resemble those of a tiger, especially in terms of ear features; '*Sunflower*' frequently induces circular patterns in the output; and '*Peacock*' consistently produces outputs where new structures emerge at the fixed position of the results (body of '*Peacock*'), and editing occurs around the wing area. In contrast, when using a more abstract pattern like '*Mandala*' as the target, the protection does not preserve the mandala pattern itself. Instead, it prevents the model from referencing the original semantics of the source image, resulting in an output that diverges significantly from the source content.

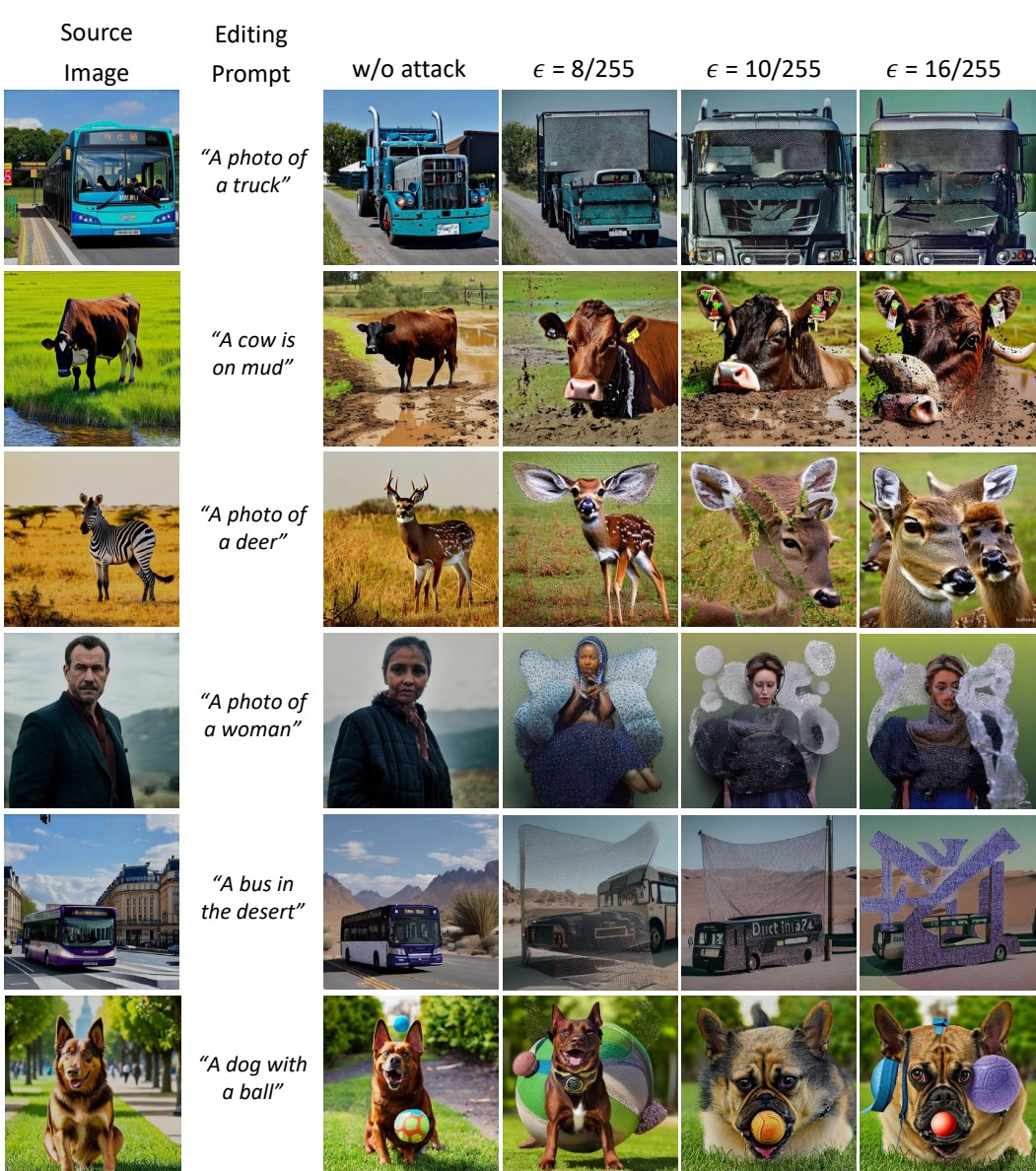

Figure 10: Qualitative comparison between different constraint parameters.

Table 16: Ablation study on the timestep set.

| Timestep Set | $K_1$ | $K_2$ | $K$ (Ours) |
|---|---|---|---|
| PSNR ↓ | 14.42 | 14.37 | 14.19 |
| SSIM ↓ | 0.354 | 0.370 | 0.332 |
| VIFp ↓ | 0.089 | 0.092 | 0.082 |
| FSIM ↓ | 0.654 | 0.652 | 0.642 |
| LPIPS ↑ | 0.600 | 0.597 | 0.606 |

## 10.2 QUALITATIVE COMPARISON WITH UNIVERSAL BASELINES

We present a more qualitative comparison with universal baselines (*i.e.* Encoder, Embedding, and Map) on U-Net-based Stable Diffusion V1.5 (Rombach et al., 2022) and DiT-based Stable Diffusion V3 (Esser et al., 2024). In Figure 12, our method consistently disrupts the reference to the original image's semantics, thereby leading to the generation of images that, regardless of the source image, primarily reflect the target semantic (e.g., tiger) during the editing process. Similar results are

Table 17: Ablation study of loss components on black-box transferability with diverse target selections. * denotes the white-box model.

(a) Ablation study for the "Ronaldo" target across different diffusion models.

| Model | Stable Diffusion V1.4 | | | Stable Diffusion V1.5* | | | Stable Diffusion V2.0 | | | InstructPix2Pix | | |
|---|---|---|---|---|---|---|---|---|---|---|---|---|
| Method | $\text{Inj}_{DF}$ | Inj | Inj+Sup | $\text{Inj}_{DF}$ | Inj | Inj+Sup | $\text{Inj}_{DF}$ | Inj | Inj+Sup | $\text{Inj}_{DF}$ | Inj | Inj+Sup |
| PSNR ↓ | 15.42 | 14.98 | **14.78** | 15.56 | 14.99 | **14.69** | 12.97 | 12.65 | **12.50** | 16.13 | 15.34 | **15.03** |
| SSIM ↓ | 0.403 | 0.388 | **0.364** | 0.447 | 0.421 | **0.383** | 0.294 | 0.288 | **0.273** | 0.530 | 0.474 | **0.452** |
| VIFp ↓ | 0.110 | 0.098 | **0.086** | 0.126 | 0.117 | **0.095** | 0.065 | 0.061 | **0.057** | 0.182 | 0.154 | **0.141** |
| FSIM ↓ | 0.682 | 0.666 | **0.657** | 0.685 | 0.677 | **0.660** | 0.597 | 0.587 | **0.581** | 0.777 | 0.737 | **0.721** |
| LPIPS ↑ | 0.535 | 0.553 | **0.576** | 0.492 | 0.541 | **0.576** | 0.605 | 0.639 | **0.649** | 0.439 | 0.482 | **0.502** |

(b) Ablation study for the "Sunflower" target across different diffusion models.

| Model | Stable Diffusion V1.4 | | | Stable Diffusion V1.5* | | | Stable Diffusion V2.0 | | | InstructPix2Pix | | |
|---|---|---|---|---|---|---|---|---|---|---|---|---|
| Method | $\text{Inj}_{DF}$ | Inj | Inj+Sup | $\text{Inj}_{DF}$ | Inj | Inj+Sup | $\text{Inj}_{DF}$ | Inj | Inj+Sup | $\text{Inj}_{DF}$ | Inj | Inj+Sup |
| PSNR ↓ | 14.91 | 14.68 | **14.43** | 14.95 | 14.60 | **14.32** | 12.61 | 12.57 | **12.35** | 16.22 | 15.96 | **15.42** |
| SSIM ↓ | 0.367 | 0.358 | **0.334** | 0.413 | 0.369 | **0.340** | 0.284 | 0.283 | **0.267** | 0.495 | 0.476 | **0.454** |
| VIFp ↓ | 0.098 | 0.094 | **0.083** | 0.117 | 0.103 | **0.088** | 0.064 | 0.063 | **0.058** | 0.179 | 0.168 | **0.158** |
| FSIM ↓ | 0.666 | 0.660 | **0.651** | 0.679 | 0.672 | **0.651** | 0.588 | 0.586 | **0.581** | 0.776 | 0.764 | **0.742** |
| LPIPS ↑ | 0.553 | 0.572 | **0.599** | 0.544 | 0.561 | **0.610** | 0.648 | 0.643 | **0.660** | 0.449 | 0.485 | **0.508** |

(c) Ablation study for the "Peacock" target across different diffusion models.

| Model | Stable Diffusion V1.4 | | | Stable Diffusion V1.5* | | | Stable Diffusion V2.0 | | | InstructPix2Pix | | |
|---|---|---|---|---|---|---|---|---|---|---|---|---|
| Method | $\text{Inj}_{DF}$ | Inj | Inj+Sup | $\text{Inj}_{DF}$ | Inj | Inj+Sup | $\text{Inj}_{DF}$ | Inj | Inj+Sup | $\text{Inj}_{DF}$ | Inj | Inj+Sup |
| PSNR ↓ | 14.58 | 14.45 | **14.21** | 14.75 | 14.26 | **14.22** | 12.36 | 12.28 | **12.06** | 16.11 | 15.58 | **15.39** |
| SSIM ↓ | 0.308 | 0.304 | **0.284** | 0.356 | 0.339 | **0.308** | 0.243 | 0.244 | **0.226** | 0.460 | 0.430 | **0.420** |
| VIFp ↓ | 0.090 | 0.087 | **0.079** | 0.106 | **0.083** | 0.088 | 0.059 | 0.058 | **0.054** | 0.166 | 0.160 | **0.155** |
| FSIM ↓ | 0.646 | 0.643 | **0.634** | 0.662 | 0.653 | **0.651** | 0.572 | 0.573 | **0.563** | 0.773 | 0.759 | **0.754** |
| LPIPS ↑ | 0.562 | 0.589 | **0.608** | 0.548 | 0.608 | **0.610** | 0.645 | 0.661 | **0.673** | 0.477 | 0.502 | **0.512** |

(d) Ablation study for the "Mandala" target across different diffusion models.

| Model | Stable Diffusion V1.4 | | | Stable Diffusion V1.5* | | | Stable Diffusion V2.0 | | | InstructPix2Pix | | |
|---|---|---|---|---|---|---|---|---|---|---|---|---|
| Method | $\text{Inj}_{DF}$ | Inj | Inj+Sup | $\text{Inj}_{DF}$ | Inj | Inj+Sup | $\text{Inj}_{DF}$ | Inj | Inj+Sup | $\text{Inj}_{DF}$ | Inj | Inj+Sup |
| PSNR ↓ | 14.79 | 14.40 | **14.19** | 15.20 | 14.89 | **14.25** | 12.48 | 12.21 | **12.17** | 15.94 | 15.31 | **15.12** |
| SSIM ↓ | 0.337 | 0.302 | **0.279** | 0.342 | 0.313 | **0.301** | 0.249 | 0.226 | **0.215** | 0.476 | 0.409 | **0.393** |
| VIFp ↓ | 0.088 | 0.074 | **0.068** | 0.092 | **0.077** | 0.078 | 0.055 | 0.050 | **0.049** | 0.166 | 0.143 | **0.136** |
| FSIM ↓ | 0.658 | 0.639 | **0.634** | 0.651 | 0.645 | **0.641** | 0.576 | **0.562** | 0.564 | 0.769 | 0.735 | **0.723** |
| LPIPS ↑ | 0.558 | 0.602 | **0.621** | 0.587 | 0.615 | **0.621** | 0.641 | 0.665 | **0.674** | 0.455 | 0.517 | **0.534** |

observed with the DiT-based Stable Diffusion V3 in Figure 13, providing visual evidence that our method achieves more robust cross-architecture transferability than existing methods.

## 10.3 QUALITATIVE COMPARISON WITH IMAGE-SPECIFIC BASELINES

We provide additional image manipulation results compared with image-specific baselines, EA, DA (Salman et al., 2023), and SA (Lo et al., 2024), using a U-Net-based (Stable Diffusion V1.5 (Rombach et al., 2022)) and DiT-based (Stable Diffusion V3 (Esser et al., 2024))generative models. In the experiments, we use the UAP generated with '*tiger*' as the target concept. In Figure 14 and 15, our method provides visually reliable image protection compared to image-specific immunization approaches on both U-Net and DiT-based image manipulation. Consistent with our motivation that an editing process which does not reference the original image cannot be considered a successful edit, this tendency is even more evident when compared against image-specific methods. We believe that these results stem from the robustness of our targeted UAP, which drives semantic shifts more strongly than existing methods.

## 10.4 QUALITATIVE RESULTS FOR IMAGE INPAINTING

To assess the generalization of our method, we additionally perform qualitative evaluations on both U-Net– and DiT-based inpainting models, using Stable Diffusion V1.5 (Rombach et al., 2022) and FLUX (Labs, 2024), respectively, with the target concept '*tiger*,' consistent with the image editing experiments. DG and AP denote DiffusionGuard (Choi et al., 2025) and AdvPaint (Jeon et al., 2025), respectively, both specifically designed for inpainting immunization. As shown in Figure 16

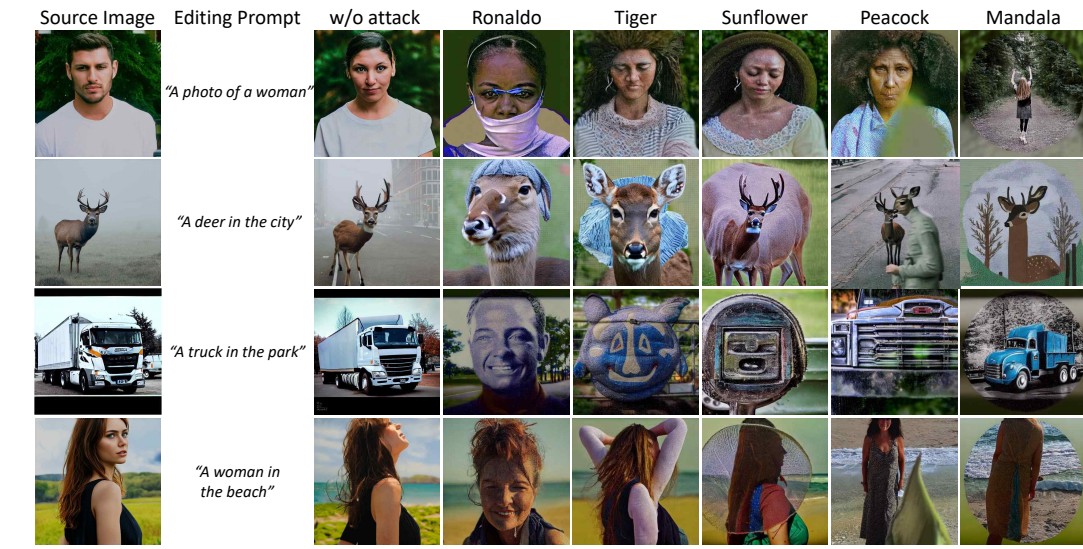

| Source Image | Editing Prompt | w/o attack | Ronaldo | Tiger | Sunflower | Peacock | Mandala |

Figure 11: Qualitative results for diverse target contents.

and 17, although our method is not tailored for inpainting, it still achieves visually competitive results compared to AP and DG. Specifically, in the U-Net–based model (Figure 16), the inpainting model interprets the absence of the object and consequently generates a new instance based solely on the prompt. In contrast, in the DiT-based model (Figure 17), while a new target object in the editing prompt is not produced, our method induces modifications that deviate more substantially from the original editing outputs than those of other methods. These observations suggest that our approach holds promise for extension as a defense mechanism against malicious inpainting.

### 10.5 QUALITY OF IMMUNIZED IMAGES

We qualitatively compare the immunized images produced by prior image-specific methods with those defended using our UAP. As shown in Figure 18, the immunized images generated by our UAP exhibit artifacts that are comparably imperceptible to those produced by image-specific methods, even across diverse domains such as real-world and paintings This observation is consistent with the high imperceptibility scores obtained in our human evaluation in Table 13 and further highlights the practicality of our universal approach to real-world application.

### 10.6 FAILURE CASES

Figure 19 illustrates several failure cases of our method. In particular, when the protected image shares similar shapes or characteristics with the target semantics embedded by the UAP, our approach may fail to effectively block editing attempts, as shown in Figure 19a. Nevertheless, this limitation can be mitigated by selectively choosing diverse targets, and our ablation study on diverse targets (see in Sec. C.3 and Sec. D.1) supports this claim. Moreover, as shown in Figure 19b, our method occasionally does not prevent undesired inpainting. While it performs well in most image-to-image editing scenarios, it exhibits limitations in inpainting, particularly when the editing region is excessively large or extremely small relative to the overall image size. In contrast, inpainting-specific baselines such as DG and AP, which are trained with access to human-provided masks or masks explicitly generated by GroundedSAM, more effectively suppress such edits. Further discussion of these failure cases is provided in Sec. 11.

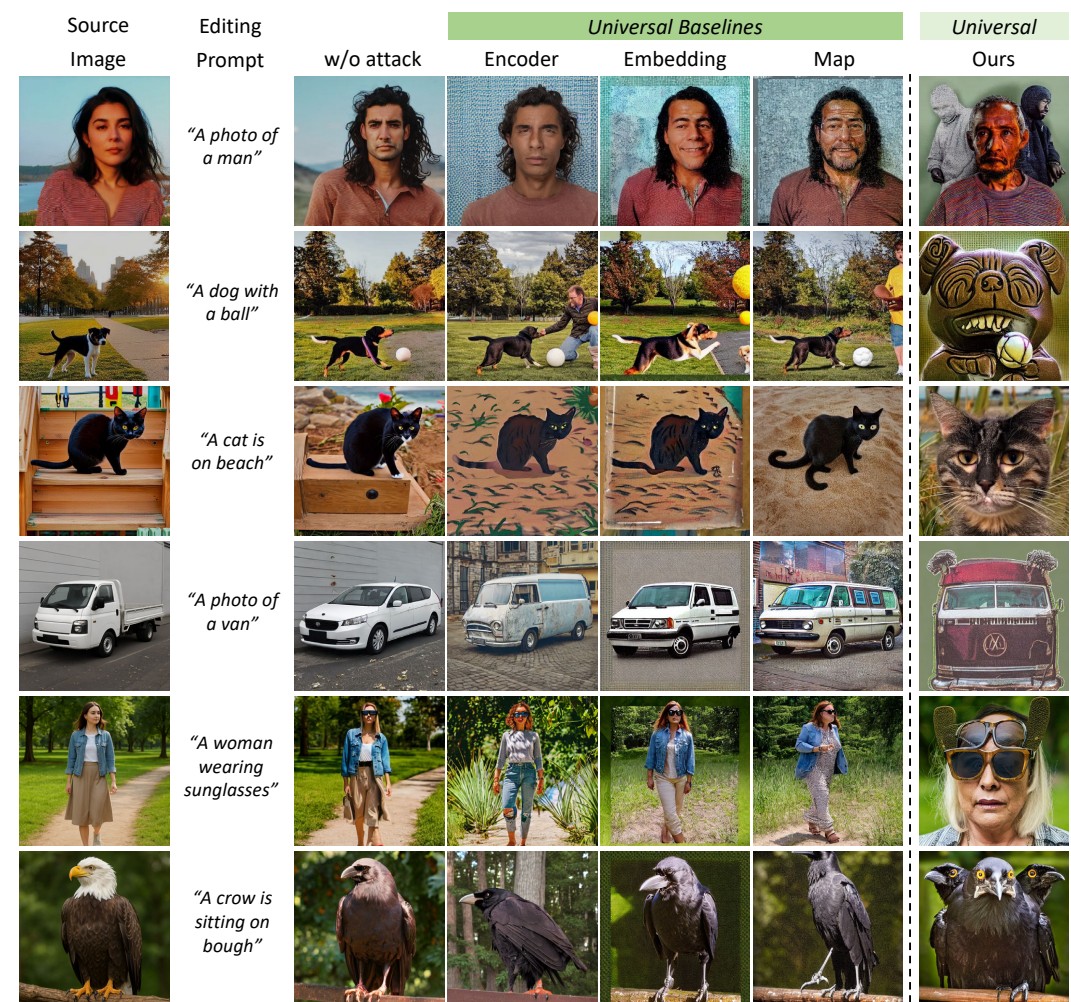

Figure 12: Qualitative comparison with universal baselines on Stable Diffusion V1.5.

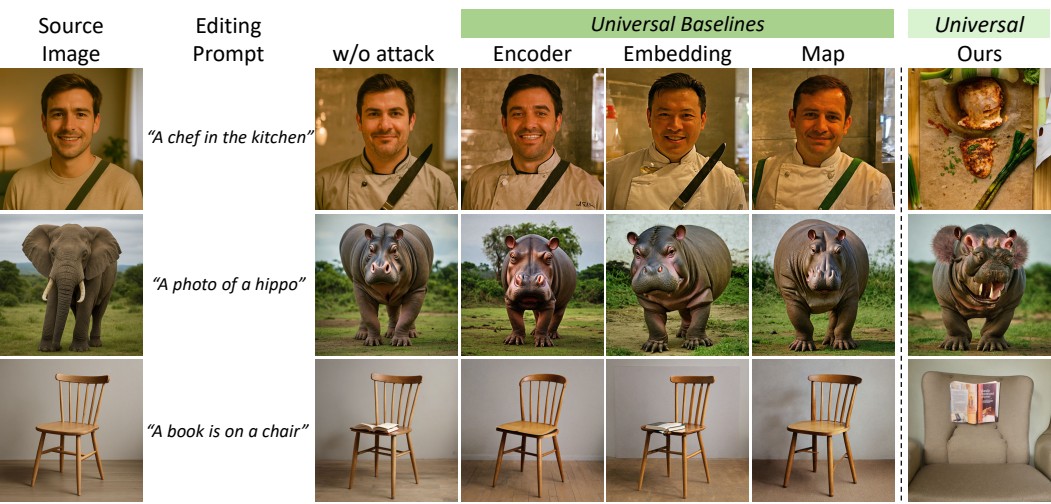

Figure 13: Qualitative comparison for image editing with universal baselines on Stable Diffusion V3 (Esser et al., 2024). The target of UAP is '*tiger*.'

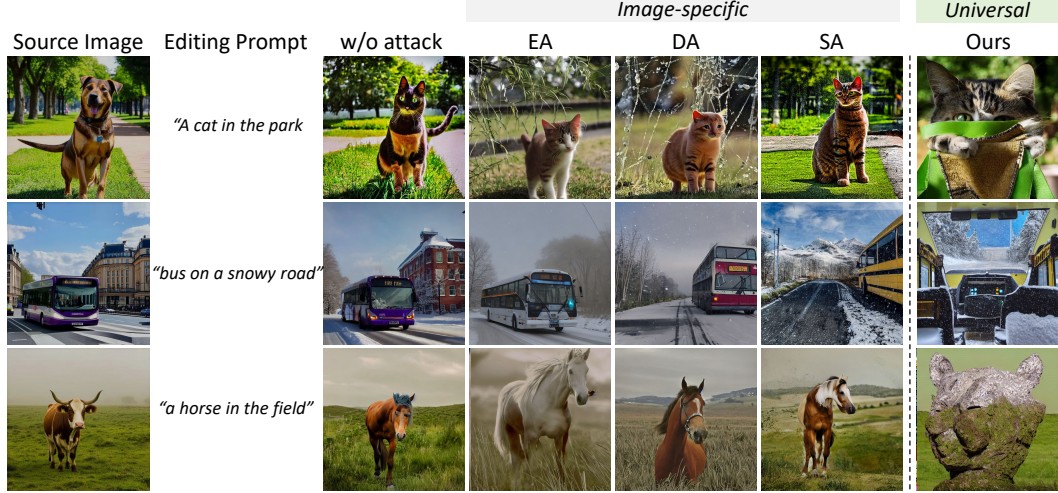

Figure 14: Qualitative comparison for image editing with image-specific baselines on Stable Diffusion V1.5 (Rombach et al., 2022). The target of UAP is '*tiger*.'

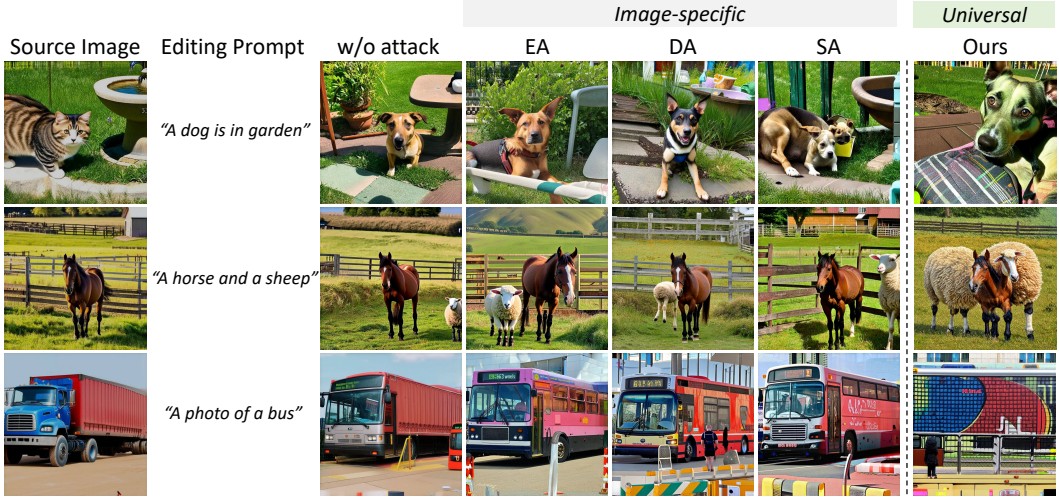

Figure 15: Qualitative comparison for image editing with image-specific baselines for image editing with Stable Diffusion V3 (Esser et al., 2024). The target is '*tiger*.'

## 11 LIMITATIONS AND DISCUSSIONS

Despite the strong qualitative and quantitative results, our method encounters limitations in certain cases. One such limitation lies in its universality: unlike existing immunization methods that produce image-specific perturbations, our UAP is a single, input-agnostic perturbation shared across all images. Consequently, when the structure or shape of an input image closely resembles that of the target images used to train the UAP (see in Figure 5 and 19a), semantic injection may fail, resulting in unsuccessful prevention of malicious editing. This limitation can be easily mitigated by using different target prompts, or, even under the same target prompt, by employing multiple target images (see in Figure 6).

Another limitation arises in inpainting scenarios. As shown in Figure 19b, when the inpainting mask is excessively large (*i.e.* remaining region is too small), the method struggles to fully prevent undesired edits, since a large portion of the UAP may be removed or it becomes difficult to appropriately fill our target semantics. This limitation is partly due to the fact that our approach is originally designed for image editing rather than inpainting, whereas inpainting-specific methods like AP (Jeon

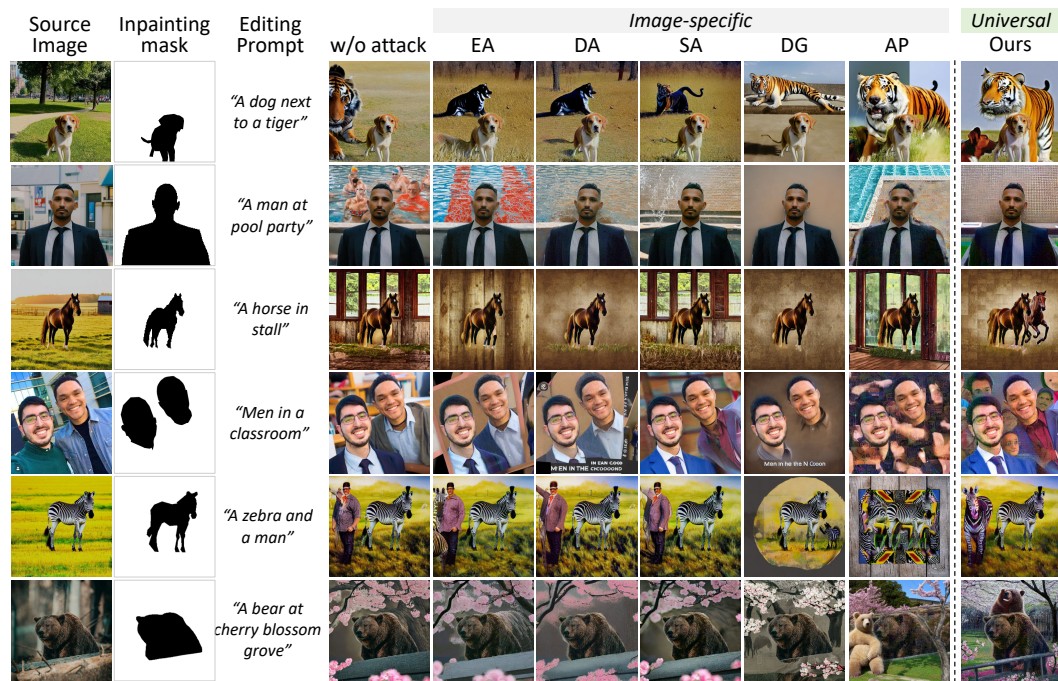

Figure 16: Qualitative results for image inpainting with Stable Diffusion V1.5 (Rombach et al., 2022). The inpainting is applied to the white regions of the mask and the target of UAP is '*tiger*.'

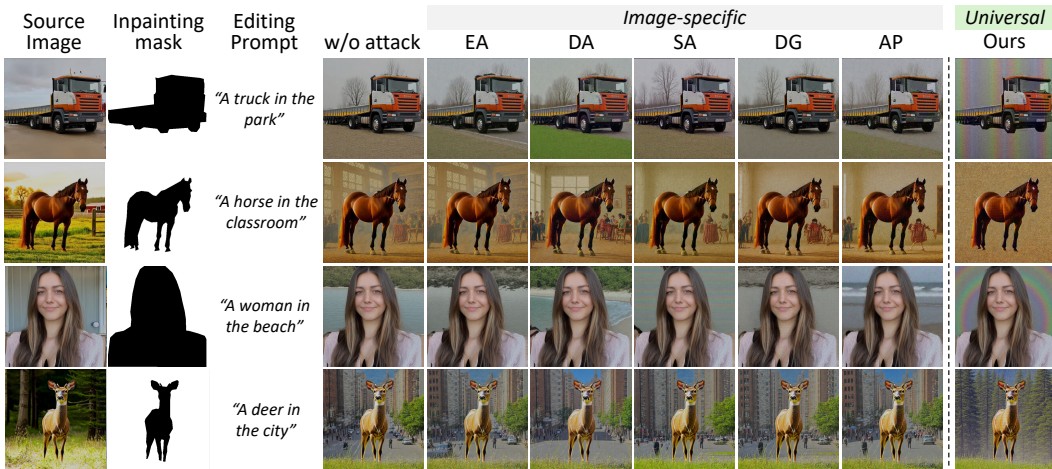

Figure 17: Qualitative results for image inpainting with Flux (Labs, 2024). White regions in the mask indicate areas to be inpainted. The target of UAP is '*tiger*.'

et al., 2025) and DG (Choi et al., 2025) explicitly incorporate human-provided masks or masks explicitly generated by GroundedSAM during the training of their perturbations, resulting in stronger performance for such tasks. However, this phenomenon is not unique to our method, as similar failures are observed in other approaches like EA, DA (Salman et al., 2023), and SA (Lo et al., 2024), which also do not leverage masks during training.

## 12 SOCIETAL IMPACTS

Our work introduces a universal adversarial perturbation (UAP) that effectively immunizes images against a wide range of diffusion-based manipulations using a single perturbation. This lightweight, data-agnostic, and generalizable defense can help protect users from malicious image editing or

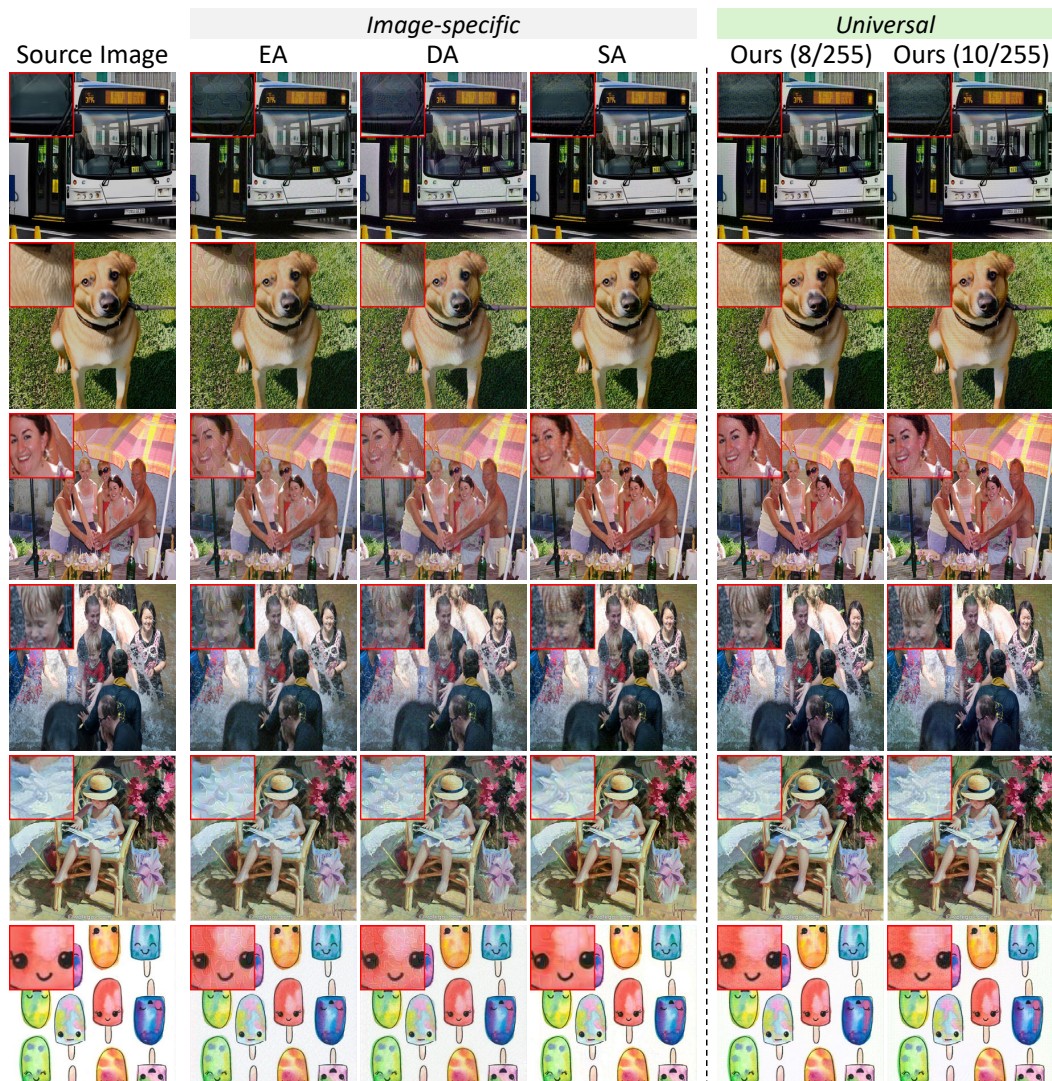

Figure 18: Comparison of immunized images produced by our method and previous image-specific approaches. Ours (8/255) and Ours (10/255) denote our method applied with $\epsilon = 8/255$ and $\epsilon = 10/255$ respectively. The target of UAP is '*tiger*.'

content forgery, particularly in cases where unauthorized modifications may cause misinformation, reputational damage, or social harm.

However, we also recognize the potential for misuse. Since our approach manipulates the behavior of generative models through adversarial signals, similar techniques could be exploited to bypass safety filters, induce unintended outputs, or interfere with intended ones. In particular, targeted UAPs could be reverse-engineered to maliciously exploit vulnerabilities in diffusion models.

e

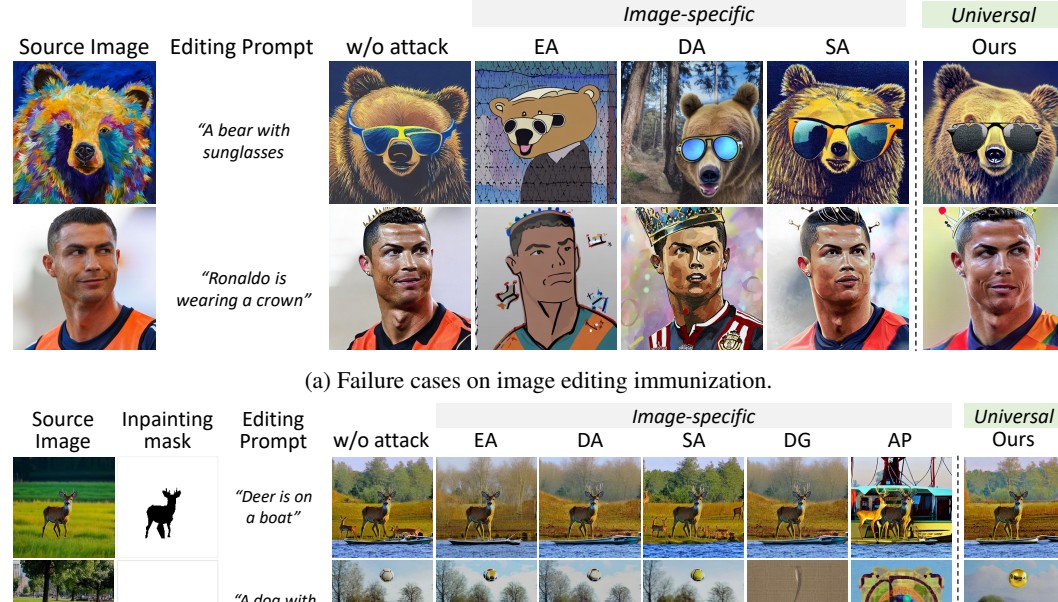

(a) Failure cases on image editing immunization.

(b) Failure cases on image inpainting immunization.

Figure 19: Visualization of failure cases. (a) The first row shows the target '*tiger*', while the second row shows the target '*ronaldo*'. (b) The white regions in the mask indicate areas where the generative model synthesizes new content.

Table 18: White-box evaluation with complex editing prompts. *Ours* denotes the full method. The best result is shown in bold; the second-best is underlined.

| Method | Clean | Universal Baselines | | | Ours |
|---|---|---|---|---|---|
| | | Enc. | Emb. | Map | |
| PSNR ↓ | – | 17.73 | 16.70 | 17.19 | **15.66** |
| SSIM ↓ | – | 0.608 | 0.447 | 0.562 | **0.422** |
| VIFp ↓ | – | 0.221 | 0.148 | 0.192 | **0.121** |
| FSIM ↓ | – | 0.774 | 0.720 | 0.747 | **0.687** |
| LPIPS ↑ | – | 0.369 | 0.511 | 0.417 | **0.548** |
| Feat. Sim. (CLIP) ↓ | 0.744 | 0.712 | 0.703 | 0.707 | **0.689** |
| Feat. Sim. (DINO) ↓ | 0.533 | 0.423 | 0.426 | 0.408 | **0.351** |

Table 19: Performance comparison across different numbers of training samples. Best results are in bold, second-best are underlined.

| Metric | Clean | Image-specific | | | | Universal | | | | |
|---|---|---|---|---|---|---|---|---|---|---|
| | | EA | DA | SA | ES | #10 | #100 | #1000 | #10000 | Ours_DF |
| PSNR ↓ | – | 14.75 | 14.71 | 16.28 | 16.04 | 15.14 | 14.52 | 14.37 | **14.19** | 14.68 |
| SSIM ↓ | – | 0.386 | 0.382 | 0.431 | 0.335 | 0.408 | 0.373 | 0.359 | **0.332** | 0.378 |
| VIFp ↓ | – | 0.088 | 0.106 | 0.138 | 0.106 | 0.124 | 0.100 | 0.093 | **0.082** | 0.106 |
| FSIM ↓ | – | **0.637** | 0.660 | 0.714 | 0.691 | 0.681 | 0.658 | 0.652 | 0.642 | 0.666 |
| LPIPS ↑ | – | 0.584 | 0.552 | 0.490 | 0.553 | 0.517 | 0.564 | 0.575 | **0.606** | 0.557 |
| Feat. Sim. (CLIP) ↓ | 0.744 | 0.677 | **0.662** | 0.705 | 0.684 | 0.696 | 0.686 | 0.682 | 0.673 | 0.685 |
| Feat. Sim. (DINO) ↓ | 0.534 | 0.373 | **0.333** | 0.447 | 0.387 | 0.402 | 0.365 | 0.360 | 0.345 | 0.376 |

Table 20: Ablation study for loss functions. Best results are shown in bold; second-best are underlined.

| Loss Function | Clean | $\mathcal{L}_{\text{sup}}$ | $\mathcal{L}_{\text{inj}}$ | $\mathcal{L}_{\text{inj}} + \mathcal{L}_{\text{sup}}$ |
|---|---|---|---|---|
| PSNR ↓ | – | 16.11 | 14.41 | **14.19** |
| SSIM ↓ | – | 0.420 | 0.367 | **0.332** |
| VIFp ↓ | – | 0.140 | 0.096 | **0.082** |
| FSIM ↓ | – | 0.697 | 0.655 | **0.642** |
| LPIPS ↑ | – | 0.490 | 0.585 | **0.606** |
| Feat. Sim. (CLIP) ↓ | 0.744 | 0.688 | 0.680 | **0.673** |
| Feat. Sim. (DINO) ↓ | 0.534 | 0.415 | 0.372 | **0.345** |

Table 21: Comparison between using attention map and cross-attention with different loss functions. Best results are shown in bold; second-best are underlined.

| Method | Clean | $\text{Map}_{sup}$ | $\text{Map}_{inj}$ | $\text{Map}_{inj+sup}$ | $\text{CA}_{sup}$ | $\text{CA}_{inj}$ | $\text{CA}_{inj+sup}$ |
|---|---|---|---|---|---|---|---|
| PSNR ↓ | – | 16.16 | 15.59 | 15.45 | 16.11 | 14.41 | **14.19** |
| SSIM ↓ | – | 0.468 | 0.358 | 0.335 | 0.420 | 0.367 | **0.332** |
| VIFp ↓ | – | 0.152 | 0.121 | 0.115 | 0.140 | 0.096 | **0.082** |
| FSIM ↓ | – | 0.710 | 0.686 | 0.679 | 0.697 | 0.655 | **0.642** |
| LPIPS ↑ | – | 0.465 | 0.537 | 0.550 | 0.490 | 0.585 | **0.606** |
| Feat. Sim. (CLIP) ↓ | 0.744 | 0.708 | 0.702 | 0.696 | 0.688 | 0.680 | **0.673** |
| Feat. Sim. (DINO) ↓ | 0.534 | 0.438 | 0.414 | 0.405 | 0.415 | 0.372 | **0.345** |

Table 22: Comparison of different data sources for source semantic suppression loss. Best results are shown in bold; second-best are underlined.

| Method | Clean | $\text{Laion}_{sup}$ | $\text{ImageNet-Excluded}_{sup}$ |
|---|---|---|---|
| PSNR ↓ | – | 16.11 | **16.06** |
| SSIM ↓ | – | **0.420** | 0.440 |
| VIFp ↓ | – | **0.140** | 0.142 |
| FSIM ↓ | – | **0.697** | 0.700 |
| LPIPS ↑ | – | **0.490** | 0.485 |
| Feat. Sim. (CLIP) ↓ | 0.744 | **0.688** | 0.699 |
| Feat. Sim. (DINO) ↓ | 0.534 | **0.415** | 0.421 |

Table 23: Immunization performance of checkerboard target image. * indicates the white-box result.

| Model | | Stable Diffusion V1.4 | | | | | Stable Diffusion V1.5* | | | | | Stable Diffusion V2.0 | | | | | InstructPix2Pix | | | |
|---|---|---|---|---|---|---|---|---|---|---|---|---|---|---|---|---|---|---|---|---|
| Method | clean | Enc. | Emb. | Map | Ours | clean | Enc. | Emb. | Map | Ours | clean | Enc. | Emb. | Map | Ours | clean | Enc. | Emb. | Map | Ours |
| PSNR ↓ | – | 18.08 | 17.00 | 17.51 | **16.00** | – | 16.55 | 15.80 | 16.16 | **15.19** | – | 14.33 | 14.25 | 13.89 | **13.54** | – | 18.18 | 17.11 | 17.12 | **15.70** |
| SSIM ↓ | – | 0.634 | 0.467 | 0.584 | **0.315** | – | 0.482 | 0.378 | 0.468 | **0.263** | – | 0.417 | 0.335 | 0.355 | **0.225** | – | 0.571 | 0.521 | 0.498 | **0.399** |
| VIFp ↓ | – | 0.239 | 0.162 | 0.210 | **0.091** | – | 0.154 | 0.117 | 0.152 | **0.070** | – | 0.122 | 0.099 | 0.107 | **0.067** | – | 0.205 | 0.207 | 0.204 | **0.134** |
| FSIM ↓ | – | 0.787 | 0.732 | 0.759 | **0.679** | – | 0.714 | 0.689 | 0.710 | **0.654** | – | 0.654 | 0.644 | 0.624 | **0.594** | – | 0.816 | 0.790 | 0.805 | **0.743** |
| LPIPS ↑ | – | 0.348 | 0.492 | 0.399 | **0.591** | – | 0.452 | 0.548 | 0.465 | **0.618** | – | 0.510 | 0.579 | 0.556 | **0.631** | – | 0.372 | 0.419 | 0.442 | **0.520** |
| Feat. Sim. (C) ↓ | 0.743 | 0.707 | 0.690 | 0.702 | **0.653** | 0.743 | 0.708 | 0.696 | 0.704 | **0.658** | 0.694 | 0.652 | 0.647 | 0.643 | **0.620** | 0.810 | 0.787 | 0.756 | 0.769 | **0.746** |
| Feat. Sim. (D) ↓ | 0.531 | 0.436 | 0.399 | 0.421 | **0.334** | 0.531 | 0.438 | 0.436 | 0.424 | **0.338** | 0.408 | 0.275 | 0.274 | 0.257 | **0.212** | 0.658 | 0.646 | 0.590 | 0.647 | **0.580** |

Table 24: White-box immunization performance on Stable Diffusion V1.5 compared with image-specific methods. $\text{Ours}_{10}$ and $\text{Ours}_{16}$ correspond to using a universal perturbation budget of 10/255 and an image-specific perturbation budget of 16/255, respectively.

| Method | clean | EA | DA | SA | ES | $\text{Ours}_{10}$ | $\text{Ours}_{16}$ |
|---|---|---|---|---|---|---|---|
| PSNR ↓ | – | 14.75 | 14.71 | 16.28 | 16.04 | 14.19 | **13.31** |
| SSIM ↓ | – | 0.386 | 0.382 | 0.431 | 0.335 | 0.332 | **0.277** |
| VIFp ↓ | – | 0.088 | 0.106 | 0.138 | 0.106 | 0.082 | **0.061** |
| FSIM ↓ | – | 0.637 | 0.660 | 0.714 | 0.691 | 0.642 | **0.612** |
| LPIPS ↑ | – | 0.584 | 0.552 | 0.490 | 0.553 | 0.606 | **0.677** |
| Feat. Sim. (C) ↓ | 0.744 | 0.677 | 0.662 | 0.705 | 0.684 | 0.670 | **0.658** |
| Feat. Sim. (D) ↓ | 0.534 | 0.373 | 0.333 | 0.447 | 0.387 | 0.345 | **0.295** |

Table 25: Black-box transferability on SD V1.4, SD V2.0, and InstructPix2Pix. $\text{Ours}_{10}$ and $\text{Ours}_{16}$ correspond to using a universal perturbation budget of 10/255 and an image-specific perturbation budget of 16/255, respectively. * indicates the white-box result.

| Model | | Stable Diffusion V1.4 | | | | | | Stable Diffusion V2.0 | | | | | | InstructPix2Pix | | | | | |
|---|---|---|---|---|---|---|---|---|---|---|---|---|---|---|---|---|---|---|---|
| Method | Clean | EA | DA | SA | ES | $\text{Ours}_{10}$ | $\text{Ours}_{16}$ | Clean | EA | DA | SA | ES | $\text{Ours}_{10}$ | $\text{Ours}_{16}$ | Clean | EA | DA | SA | ES | $\text{Ours}_{10}$ | $\text{Ours}_{16}$ |
| PSNR ↓ | – | 15.46 | 15.54 | 14.91 | 16.24 | 14.18 | **12.73** | – | 12.48 | 12.73 | 12.79 | 13.74 | 12.17 | **11.05** | – | 16.02 | 15.89 | 17.26 | 14.77 | 15.36 | **13.62** |
| SSIM ↓ | – | 0.444 | 0.457 | 0.336 | 0.408 | 0.332 | **0.232** | – | 0.298 | 0.304 | 0.271 | 0.317 | 0.240 | **0.182** | – | 0.445 | 0.409 | 0.499 | 0.396 | 0.418 | **0.348** |
| VIFp ↓ | – | 0.119 | 0.143 | 0.088 | 0.141 | 0.083 | **0.042** | – | 0.074 | 0.085 | 0.060 | 0.092 | 0.057 | **0.035** | – | 0.133 | 0.134 | 0.189 | 0.128 | 0.135 | **0.105** |
| FSIM ↓ | – | 0.664 | 0.695 | 0.666 | 0.691 | 0.642 | **0.591** | – | 0.558 | 0.586 | 0.632 | 0.608 | 0.555 | **0.529** | – | 0.747 | 0.755 | 0.795 | 0.702 | 0.713 | **0.681** |
| LPIPS ↑ | – | 0.545 | 0.507 | 0.549 | 0.510 | 0.604 | **0.701** | – | 0.634 | 0.611 | 0.626 | 0.593 | 0.660 | **0.721** | – | 0.509 | 0.533 | 0.433 | 0.529 | 0.527 | **0.587** |
| Feat. Sim. (C) ↓ | 0.743 | 0.663 | 0.675 | 0.701 | 0.704 | 0.675 | **0.659** | 0.694 | 0.606 | 0.628 | 0.656 | 0.651 | 0.616 | **0.601** | 0.810 | 0.774 | 0.788 | 0.773 | 0.721 | 0.729 | **0.710** |
| Feat. Sim. (D) ↓ | 0.531 | 0.335 | 0.370 | 0.445 | 0.426 | 0.346 | **0.293** | 0.408 | 0.188 | 0.210 | 0.307 | 0.303 | 0.191 | **0.159** | 0.658 | 0.655 | 0.643 | 0.632 | 0.522 | 0.527 | **0.462** |

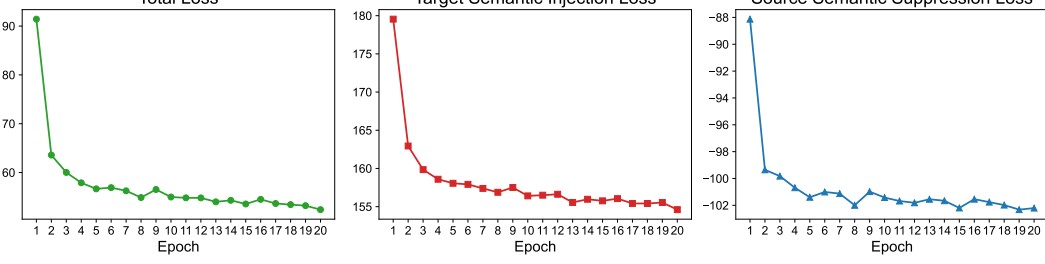

Figure 20: Visualization of loss values across training epochs.

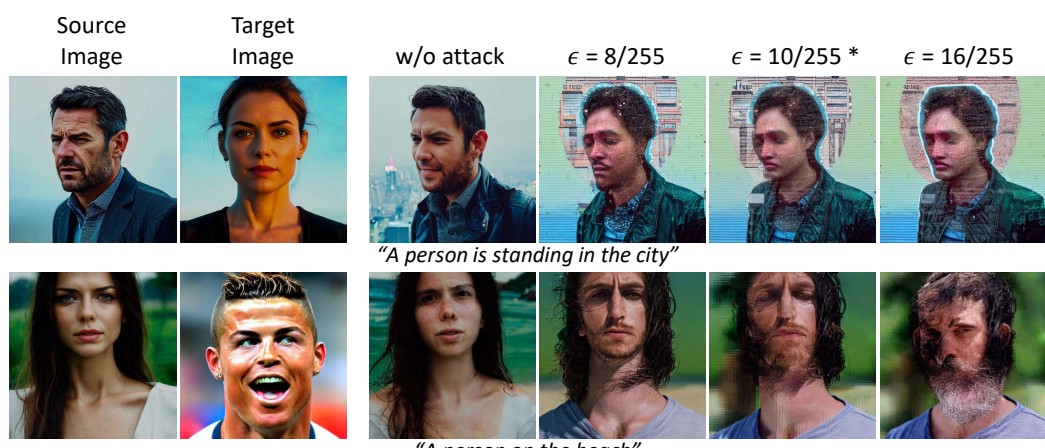

Figure 21: Ablation studies of different target identity.

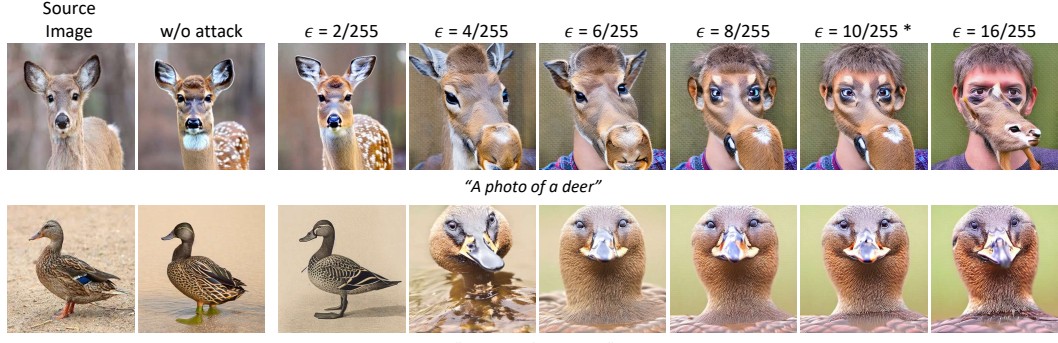

Figure 22: Visualization of attribute transferability within the **different** category between target and source images. Female deer do not have antlers, whereas male deer do. Female mallards have brown head feathers, whereas males display iridescent greenish-blue heads. UAP is generated with "Ronaldo" target.

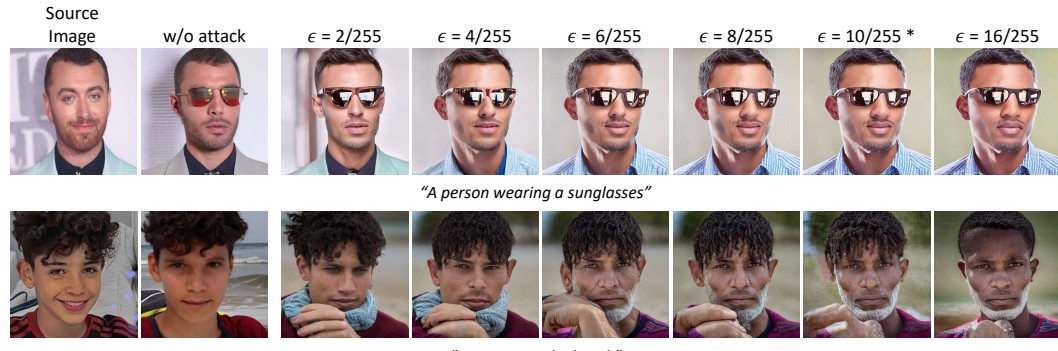

Figure 23: Visualization of attribute transferability within the **same** category between target and source images. UAP is generated with "Ronaldo" target.

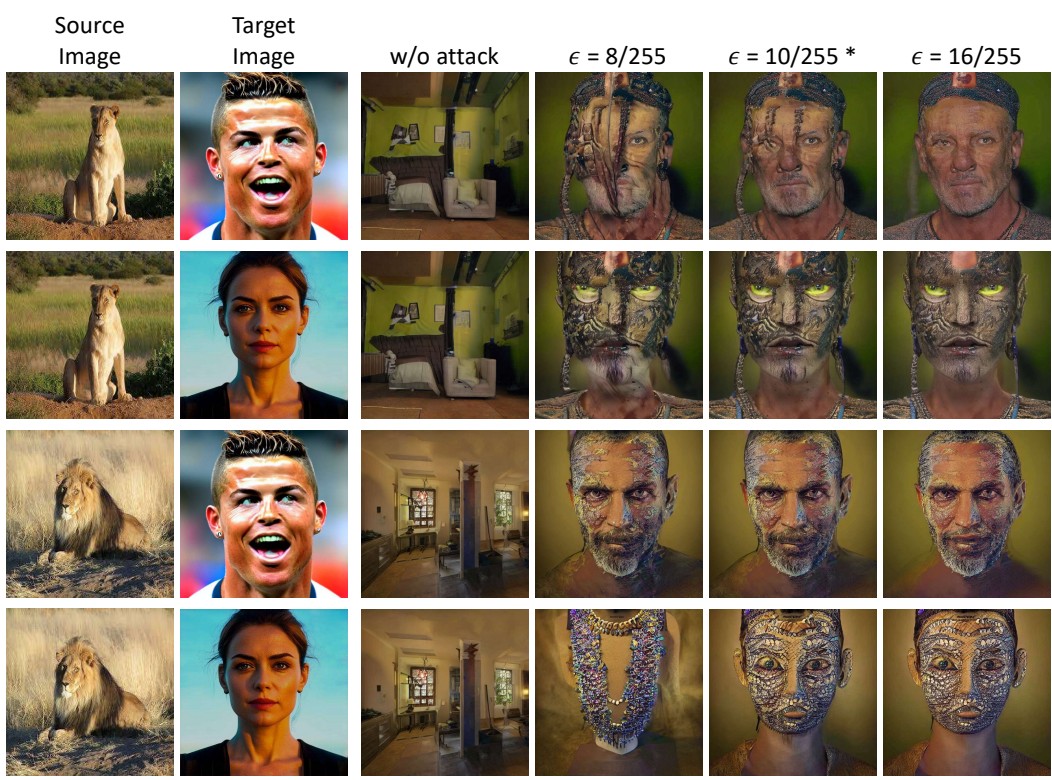

Figure 24: Qualitative results of diffusion denoising process without text condition.

