# OpenReview forum: "Universal Image Immunization against Diffusion-based Image Editing via Semantic Injection"
_ICLR.cc/2026/Conference — Submitted to ICLR 2026_

### Official Review · Reviewer_Hbgg · 2025-10-17

**Soundness:** 4
**Presentation:** 3
**Contribution:** 3
**Rating:** 8
**Confidence:** 3

**Summary:**

The paper introduces a universal adversarial perturbation approach for protecting images from diffusion-based editing. Instead of optimizing perturbations per image, it learns a single perturbation that can be applied universally. The method combines a semantic injection loss that aligns perturbed images with a target concept and a suppression loss that reduces the influence of original semantics, effectively disrupting unauthorized edits. Experiments show strong protection, cross-model generalization, and competitive performance even in data-free settings.

**Strengths:**

1. The paper is clearly written and well organized, with intuitive figures and a logical presentation of ideas.

2.  The proposed framework is well motivated, and the introduction of the source semantic suppression loss is a novel and insightful component that strengthens the overall approach.

3. The experiments are thorough and convincing, showing strong and consistent results across models and settings, including data-free and black-box scenarios.

**Weaknesses:**

The comparison with *Semantic Attack* may not be fully fair, as the original method is not designed under any $ \ell_2 $ or $ \ell_\infty $ perturbation constraint. Imposing such a bound changes its optimization behavior and could disadvantage it in this setting.

**Questions:**

1. Have the authors explored how the visual structure of the chosen target (for example, purely geometric or black-and-white grid patterns instead of semantic objects like “Ronaldo” or “Tiger”) affects the resulting perturbation? Such structured patterns might yield more uniform attention disruption and stronger transferability.

2. The method achieves strong performance under the universal constraint. If this constraint were relaxed—allowing limited image-specific adaptation—how might the performance change, and what strategies could further strengthen the performance in that less restricted setting?

---

> ### Author Response · Authors · 2025-11-23
>
> We sincerely appreciate the reviewer’s constructive feedback and insightful questions.
>
>
> > **W1**: The comparison with Semantic Attack may not be fully fair, as the original method is not designed under any ℓ₂  or ℓ∞  perturbation constraint. Imposing such a bound changes its optimization behavior and could disadvantage it in this setting.
>
>
> **A**: We would like to clarify that Semantic Attack inherently employs an ℓ∞–bounded perturbation. As shown in the original paper [1] (The first equation on p.4 and Line 12 in the pseudocode on p.5), the update step explicitly applies an ℓ∞ projection with the perturbation budget $\kappa$, ensuring that the perturbation remains within a fixed ℓ∞ radius throughout optimization. Therefore, constraining Semantic Attack under the same ℓ∞ budget in our evaluation ($\epsilon=16/255 \approx 0.06$ in [1]) is consistent with its original formulation, and we believe that this results in a fair comparison.
>
> ---
> [1] Lo et al., Distraction is all you need: Memory-efficient image immunization against diffusion-based image editing, CVPR 2024.

---

> ### Author Response · Authors · 2025-11-23
>
> > **Q1**: Have the authors explored how the visual structure of the chosen target (for example, purely geometric or black-and-white grid patterns instead of semantic objects like “Ronaldo” or “Tiger”) affects the resulting perturbation? Such structured patterns might yield more uniform attention disruption and stronger transferability.
>
> **A**: Thank you for the suggestion. In response, we additionally train a universal adversarial perturbation (UAP) using a checkerboard pattern as both the target prompt and the target image, and evaluate its effectiveness under both white-box and black-box settings.
>
> ### Table A: White-box immunization performance on Stable Diffusion v1.5 trained with checkerboard target image.
> *Best results are in **bold** and the second-best are $\underline{\text{underlined}}$.*
>
> | Metric | Clean | Enc | Emb | Map | Ours |
> |--------|--------|-----|------|------|-------|
> | **PSNR ↓** | -- | 16.55 | $\underline{15.80}$ | 16.16 | **15.19** |
> | **SSIM ↓** | -- | 0.482 | $\underline{0.378}$ | 0.468 | **0.263** |
> | **VIFp ↓** | -- | 0.154 | $\underline{0.117}$ | 0.152 | **0.070** |
> | **FSIM ↓** | -- | 0.714 | $\underline{0.689}$ | 0.710 | **0.654** |
> | **LPIPS ↑** | -- | 0.452 | $\underline{0.548}$ | 0.465 | **0.618** |
> | **Feat. Sim. (CLIP) ↓** | 0.743 | 0.708 | $\underline{0.696}$ | 0.704 | **0.658** |
> | **Feat. Sim. (DINO) ↓** | 0.531 | 0.438 | 0.436 | $\underline{0.424}$ | **0.338** |
>
>
>
>
>
>
>
>
>
> ### Table B: Black-box immunization performance on SD v1.4, SD v2.0, and InstructPix2Pix trained with checkerboard target image.
> *Best results are in **bold**, second-best are $\underline{\text{underlined}}$.*
>
> |  |  **SD 1.4** | | | | | **SD 2.0** | | | | |**IP2P** | | | | |
> |-|-|-|-|-|-|-|-|-|-|-|-|-|-|-|-|
> | Metric | Clean | Enc | Emb | Map | Ours | Clean | Enc | Emb | Map | Ours | Clean | Enc | Emb | Map | Ours |
> | **PSNR ↓** | -- | 18.08 | $\underline{17.00}$ | 17.51 | **16.00** | -- | 14.33 | 14.25 | $\underline{13.89}$ | **13.54** | -- | 18.18 | $\underline{17.11}$ | 17.12 | **15.70** |
> | **SSIM ↓** | -- | 0.634 | $\underline{0.467}$ | 0.584 | **0.315** | -- | 0.417 | $\underline{0.335}$ | 0.355 | **0.225** | -- | 0.571 | 0.521 |$\underline{0.498}$ | **0.399** |
> | **VIFp ↓** | -- | 0.239 | $\underline{0.162}$ | 0.210 | **0.091** | -- | 0.122 | $\underline{0.099}$ | 0.107 | **0.067** | -- | 0.205 | 0.207 | $\underline{0.204}$ | **0.134** |
> | **FSIM ↓** | -- | 0.787 | $\underline{0.732}$ | 0.759 | **0.679** | -- | 0.654 | 0.644 | $\underline{0.624}$ | **0.594** | -- | 0.816 | $\underline{0.790}$ | 0.805 | **0.743** |
> | **LPIPS ↑** | -- | 0.348 | $\underline{0.492}$ | 0.399 | **0.591** | -- | 0.510 | $\underline{0.579}$ | 0.556 | **0.631** | -- | 0.372 | 0.419 | $\underline{0.442}$ | **0.520** |
> | **Feat. Sim. (CLIP) ↓** | 0.743 | 0.707 | $\underline{0.690}$ | 0.702 | **0.653** | 0.694 | 0.652 | 0.647 | $\underline{0.643}$ | **0.620** | 0.810 | 0.787 | $\underline{0.756}$ | 0.769 | **0.746** |
> | **Feat. Sim. (DINO) ↓** | 0.531 | 0.436 | $\underline{0.399}$ | 0.421 | **0.334** | 0.408 | 0.275 | 0.274 | $\underline{0.257}$ | **0.212** | 0.658 | 0.646 | $\underline{0.590}$ | 0.647 | **0.580** |
>
>
> As shown in Table A and B, even with this purely geometric target, our method still outperforms other universal baselines, demonstrating that the proposed approach is agnostic to the specific form of the target concept.
>
> This experiment further highlights the robustness of our semantic injection framework, and we appreciate the suggestion that led us to include it.

---

> ### Author Response · Authors · 2025-11-23
>
> > **Q2**:The method achieves strong performance under the universal constraint. If this constraint were relaxed—allowing limited image-specific adaptation—how might the performance change, and what strategies could further strengthen the performance in that less restricted setting?
>
> **A**:Thank you for the thoughtful question. To explore how performance changes when relaxing the universal constraint, we additionally train our perturbation with ϵ=16/255, matching the setting commonly used by image-specific methods. We report the comparison against image-specific baselines in white-box setting in Table C. For a comprehensive analysis of black-box transferability, we kindly refer readers to Appendix Table 25.
>
>
> ### Table C: White-box transferability on Stable Diffusion v1.5 under image-specific perturbation budget.
> *Best results are in **bold**, second-best are $\underline{\text{underlined}}$.*
>
> | Metric | Clean | EA | DA | SA |ES| Ours₁₀ | Ours₁₆ |
> |--------|--------|------|------|------|------|---------|---------|
> | **PSNR ↓** | -- | 14.75 | 14.71 | 16.28 | 16.04 | $\underline{14.19}$ | **13.31** |
> | **SSIM ↓** | -- | 0.386 | 0.382 | 0.431 | 0.335 | $\underline{0.332}$ | **0.277** |
> | **VIFp ↓** | -- | 0.088 | 0.106 | 0.138 | 0.106 | $\underline{0.082}$ | **0.061** |
> | **FSIM ↓** | -- | $\underline{0.637}$ | 0.660 | 0.714 | 0.691 | 0.642 | **0.612** |
> | **LPIPS ↑** | -- | 0.584 | 0.552 | 0.490 | 0.553 | $\underline{0.606}$ | **0.677** |
> | **Feat. Sim (CLIP) ↓** | 0.744 | 0.677 | $\underline{0.662}$ | 0.705 | 0.684 | 0.670 | **0.658** |
> | **Feat. Sim (DINO) ↓** | 0.534 | 0.373 | $\underline{0.333}$ | 0.447 | 0.387 | 0.345 | **0.295** |
>
> Surprisingly, our method with a higher budget (ϵ=16/255) outperforms all image-specific approaches by a substantial margin under the same perturbation budget. We believe that this improvement arises because a larger perturbation budget allows the targeted UAP to enforce the target semantics more strongly, causing the model to place greater emphasis on the injected semantics and thereby more effectively disrupt the editing process.
>
> We sincerely appreciate this suggestion, as it have enabled us to demonstrate that our universal attack remains highly effective—even under the inherently more challenging universal setting—and achieves stronger immunization performance than methods that operate in an image-specific setting. This further highlights the practicality and robustness of our universal immunization strategy.

---

### Official Review · Reviewer_S8F9 · 2025-10-17

**Soundness:** 3
**Presentation:** 4
**Contribution:** 3
**Rating:** 4
**Confidence:** 4

**Summary:**

The paper presents a framework that learns a single universal adversarial perturbation (UAP) to safeguard images from unauthorized text‑guided diffusion model editing. Unlike prior approaches that rely on image‑specific perturbations—limiting scalability and practicality—the proposed method employs one universal perturbation applicable to any image. By overwriting the original semantic content with a target semantic at the cross‑attention level, the approach effectively alters the resulting edits. Experimental results demonstrate that the proposed UAP not only outperforms existing baselines in universal settings but also achieves performance comparable to image‑specific perturbations.

**Strengths:**

1. The proposed method enables universal protection using a single perturbation, making it significantly more practical and scalable compared to image-specific perturbations.
2. The paper is clearly written and easy to follow, with well-structured methodology and presentation.
3. The approach demonstrates broad applicability across diverse editing models, including Stable Diffusion v1.4 and v2.0, InstructPix2Pix, DiT, and inpainting pipelines.

**Weaknesses:**

1. While the method aims to inject target semantics, it is unclear whether the perturbation truly captures the intended concept. For instance, in the *cow* example of Figure 3 (Ours), the generated image still depicts a cow. Also, the perturbation appears to preserve only the **structure** of the *Ronaldo* target image, rather than semantic attributes like gender or identity.
2. The authors claim that text semantics are naturally fused into visual features at the cross-attention output level. However, textual semantics are also embedded within **attention map**—as used in prior works such as Lo et al. [1]—since the key vectors in the cross-attention mechanism are derived from the textual prompt. The novelty of operating on cross-attention outputs rather than attention maps may be overstated.
3. The data-dependent UAP is trained on 10,000 randomly sampled LAION-2B-en image–text pairs and evaluated on 500 images spanning 10 object classes. It remains unclear whether the semantic suppression generalizes beyond the training distribution. In particular, if a new image contains semantics absent from the 10,000 training pairs, the UAP may exhibit reduced effectiveness.
4. The proposed UAP is added to *generated* images before passing them through the diffusion model. However, this is not representative of typical editing pipelines, which usually operate on *real* images. The mismatch between training/deployment assumptions and real usage scenarios raises concerns about practical robustness.

[1] Lo et al., Distraction is all you need: Memory-efficient image immunization against diffusion-based image editing, CVPR 2024.

Note: Weaknesses 1-4 correspond directly to Questions 1-4.

**Questions:**

1. In Section 5.2, the authors claim that the generated results reflect the injected *Ronaldo* semantics. If the target image were replaced with a different individual—such as a woman or someone with distinct facial attributes—would the resulting edits reflect high-level semantic changes (e.g., gender, identity) rather than merely structural features? An expanded ablation on target identity would help assess the generality and depth of the proposed semantic injection.
2. The “Attention Map Attack” baseline generates perturbations by minimizing the alignment between the attention map and the original image semantics (Section 7.4). If this baseline were re-implemented using the same loss functions (Eq. 4 and 5), but applied to attention maps rather than cross-attention outputs, would it achieve comparable effectiveness to the proposed method? A direct comparison would clarify whether operating on cross-attention outputs offers a meaningful advantage over using attention maps.
3. How does the proposed UAP perform on test images that contain semantics not seen during training? Additional experiments would help validate the generalization of semantic suppression beyond the 10,000 training pairs.
4. All evaluations appear to apply the UAP to *generated* images. Has the method been tested on *real* images as inputs to the editing pipeline? Since most practical use cases involve real images, results on this setting would be valuable.
5. In Figure 2(b), the attention map for the *Ronaldo* prompt appears to be as responsive as that of *people*, despite the claim in the Figure 2 caption that attention should be suppressed for the target prompt. Can the authors clarify this observation?

---

> ### Author Response · Authors · 2025-11-23
>
> We sincerely appreciate the reviewer’s constructive feedback and insightful questions.
>
> > **W1/Q1**: In Section 5.2, the authors claim that the generated results reflect the injected Ronaldo semantics. If the target image were replaced with a different individual—such as a woman or someone with distinct facial attributes—would the resulting edits reflect high-level semantic changes (e.g., gender, identity) rather than merely structural features? An expanded ablation on target identity would help assess the generality and depth of the proposed semantic injection.
>
>
> **A**: Thank you for the insightful question. To examine whether our UAP truly injects high-level target semantics rather than merely structural cues, we add a new ablation in Figure 21 of the Appendix. To minimize explicit identity information provided via the prompts, we use a neutral prompt (“person”) instead of gender-specific descriptors (“man,” “woman”). We then test UAPs trained with two different target identities (Ronaldo vs. a female portrait) on source images of opposite genders.
>
> We observe that when the perturbation budget is small, the source image already shows noticeable effects—such as structural distortions or gender changes—while still retaining much of its overall layout. As the perturbation budget increases, these identity-level changes become clearer, and the generated images increasingly reflect the target identity. As the perturbation budget increases, these identity-level semantic shifts become more pronounced, and the structural traits gradually start to resemble those of the target image.
>
> These results indicate that our UAP is not simply transferring low-level appearance or hallucinating the target geometry; rather, it successfully injects high-level semantics that effectively mislead the editing process.

---

> ### Author Response · Authors · 2025-11-23
>
> > **W2/Q2**: The “Attention Map Attack” baseline generates perturbations by minimizing the alignment between the attention map and the original image semantics (Section 7.4). If this baseline were re-implemented using the same loss functions (Eq. 4 and 5), but applied to attention maps rather than cross-attention outputs, would it achieve comparable effectiveness to the proposed method? A direct comparison would clarify whether operating on cross-attention outputs offers a meaningful advantage over using attention maps.
>
>
> **A**: Thank you for the insightful suggestion. Following your feedback, we additionally re-implemented the attention-map baseline using our loss functions (Eqs. 4 and 5) and provide the results in Table A.
>
> ### Table A: Comparison between using attention map and cross-attention with different loss functions
> *Best results are in **bold**, second-best are $\underline{\text{underlined}}$.*
>
> | Metric | Clean | Map$_\text{sup}$ | Map$_\text{inj}$ | Map$_\text{inj+sup}$ | CA$_\text{sup}$ | CA$_\text{inj}$ | CA$_\text{inj+sup}$ |
> |--------|--------|---------|---------|--------------|--------|---------|-------------|
> | **PSNR ↓** | -- | 16.16 | 15.59 | 15.45 | 16.11 | $\underline{14.41}$ | **14.19** |
> | **SSIM ↓** | -- | 0.468 | 0.358 | $\underline{0.335}$ | 0.420 | 0.367 | **0.332** |
> | **VIFp ↓** | -- | 0.152 | 0.121 | 0.115 | 0.140 | $\underline{0.096}$ | **0.082** |
> | **FSIM ↓** | -- | 0.710 | 0.686 | 0.679 | 0.697 | $\underline{0.655}$ | **0.642** |
> | **LPIPS ↑** | -- | 0.465 | 0.537 | 0.550 | 0.490 | $\underline{0.585}$ | **0.606** |
> | **Feat. Sim. (CLIP) ↓** | 0.744 | 0.708 | 0.702 | 0.696 | 0.688 | $\underline{0.680}$ | **0.673** |
> | **Feat. Sim. (DINO) ↓** | 0.534 | 0.438 | 0.414 | 0.405 | 0.415 | $\underline{0.372}$ | **0.345** |
>
>
> The results show that applying the injection loss improves performance compared to the untargeted version (Map), confirming the effectiveness of our targeted strategy. Moreover, when the same losses are applied to cross-attention outputs instead of attention maps, our method achieves consistently stronger performance across overall metrics. These findings indicate that operating on cross-attention outputs provides a more robust and semantically aligned immunization mechanism.
>
> Furthermore, we have added a detailed justification for why semantic injection and suppression should operate on cross-attention outputs in Section 4.4 of the revised manuscript. We would greatly appreciate it if you could take a look at the updated discussion.

---

> ### Author Response · Authors · 2025-11-23
>
> > **W3/Q3**: How does the proposed UAP perform on test images that contain semantics not seen during training? Additional experiments would help validate the generalization of semantic suppression beyond the 10,000 training pairs.
>
>
> **A**: To evaluate the generalization capability of the source semantic suppression loss, we conduct an additional experiment using ImageNet. Specifically, we exclude all 136 classes that are semantically related to the 10 object categories used in the inference set (covering all their WordNet super- and sub-classes), and randomly sample 10,000 images from the remaining categories to train the UAP, denoted as “ImageNet-Excluded”.
>
> ### Table B: Comparison of different external data sources for support and injection strategies
>
> | Metric | Clean | ImageNet-Excluded$_\text{sup}$ | Laion$_\text{sup}$ |
> |--------|--------|--------------------|----------------|
> | **PSNR ↓** | -- | 16.06 | 16.11 |
> | **SSIM ↓** | -- |  0.440 | 0.420 |
> | **VIFp ↓** | -- |  0.142 | 0.140 |
> | **FSIM ↓** | -- | 0.700 | 0.697 |
> | **LPIPS ↑** | -- |  0.485 | 0.490 |
> | **Feat. Sim. (CLIP) ↓** | 0.744 |  0.699 | 0.688 |
> | **Feat. Sim. (DINO) ↓** | 0.534 |  0.421 | 0.415 |
>
> As shown in Table B, a UAP trained on unseen semantics (ImageNet-Excluded) achieves performance comparable to the original version trained on LAION. This demonstrates that our source semantic suppression loss generalizes well beyond the semantics exposed during training, enabling robust protection even in unseen domains.

---

> ### Author Response · Authors · 2025-11-23
>
> > **W4/Q4**: All evaluations appear to apply the UAP to generated images. Has the method been tested on real images as inputs to the editing pipeline? Since most practical use cases involve real images, results on this setting would be valuable.
>
>
> **A**:Thank you for pointing this out. We fully agree that testing on real images is important.
> We would like to clarify that we have already included experiments under this setting in the supplementary material. Specifically, we evaluate our method on real images from the MS-COCO dataset and further extend the evaluation to painting-style images from DomainNet. The qualitative results are shown in Figure 8 and 9, and the quantitative results in Table 15.
>
> As shown in Table 15, our method achieves competitive, even surpass, immunization performance compared with per-image optimization-based methods across both domains. These results demonstrate that our method effectively protects real-world and painting images even at zero-cost without using GPU, further emphasizing its strong practicality.
>
>
> > **Q5**: In Figure 2(b), the attention map for the Ronaldo prompt appears to be as responsive as that of people, despite the claim in the Figure 2 caption that attention should be suppressed for the target prompt. Can the authors clarify this observation?
>
> **A**: We apologize for any confusion. Our intention in Figure 2 was to illustrate that (b) the source image alone does not produce a meaningful attention response to the target prompt (‘’Ronaldo’’), whereas (c) the immunized image—added by a UAP containing the target semantics—produces target-related attention responses. We agree that the original caption did not fully convey this intention, so we have revised it accordingly. We kindly invite the reviewer to refer to the updated version.
> (Updated caption: (b) Source images show strong attention to their own content, but since they do not intrinsically contain the target semantics (‘Ronaldo’), they do not produce any target-aligned attention.)

---

> ### Comment · Reviewer_S8F9 · 2025-11-26
>
> Thank you for addressing most of the questions. While your responses were helpful, I am still curious about one point related to W1Q1. What happens if the source image is changed from one of a human of the opposite gender to a non-human image, such as a cow or deer as shown in Figure 2? Will the results show different semantics like identity as the target image changes from a man (e.g., Ronaldo) to a woman (e.g., a woman portrait)?

---

> > ### Author Response · Authors · 2025-11-29
> >
> > Dear reviewer S8F9,
> >
> > Thank you for the insightful question. The previously shown gender-change effect—where the source and target images have opposite genders—demonstrates that our UAP encodes not only structural cues but also identity-level information from the target image.
> >
> > In response, we applied a male-human (Ronaldo)–based UAP to female animals whose species names provide no linguistic gender cues (i.e., “deer” and “mallard”). The results are provided in Figure 22 in the appendix, and we kindly invite you to take a look. Unlike the human-to-human case, the animal’s gender does not change, as humans and animals do not share a compatible gender attribute space, and such abstract concepts cannot be instantiated from a single target image. This highlights the inherent difficulty of representing cross-category identity concepts such as “gender”, within our current framework.
> >
> > Nonetheless, this opens an intriguing direction. Understanding how diffusion models encode and transfer such identity-level attributes may reveal latent biases (e.g., gender bias in generative behavior) and provide a foundation for future work on model trustworthiness. Thank you for inspiring this perspective.
> >
> > Further inspired by your question, we also examined what other attributes may transfer by conducting additional experiments on age and skin tone. As shown in Figure 23, increasing ε produces progressive tanning or causes a young boy to appear older. These results indicate that our UAP carries a broader set of target-image attributes beyond gender and that these attributes can indeed be transferred.
> >
> > Your question has greatly helped us better understand, in a more comprehensive manner, the range of semantics and features that our method actually injects.

---

### Official Review · Reviewer_8CoU · 2025-10-19

**Soundness:** 2
**Presentation:** 3
**Contribution:** 2
**Rating:** 4
**Confidence:** 4

**Summary:**

The paper empirically proposes a universal image immunization method against diffusion-based editing by jointly optimizing semantic injection and semantic suppression losses. A single universal perturbation is trained to mislead diffusion models semantically while preserving visual quality. Extensive experiments demonstrate strong white-box and black-box defense across multiple diffusion models.

**Strengths:**

- The paper proposes a universal, data-free image immunization framework that generalizes across diffusion models.
- The method introduces a simple yet effective dual-loss design to achieve semantic-level defense.
- The approach demonstrates strong transferability and robustness under both white-box and black-box settings.
- The experiments cover multiple diffusion models and editing scenarios, showing consistent performance.

**Weaknesses:**

- The paper presents an empirical approach with limited theoretical justification.
- The authors do not provide a thorough discussion on why $\mathcal{L}_\text{inj}$ is effective in the cross-attention feature space or its theoretical justification, relying instead primarily on empirical validation.
- The evaluation relies heavily on pixel and perceptual similarity metrics, despite the method's core focus on semantic injection and suppression; adding CLIPScore or Grounding DINO detection would better assess semantic alignment.
- The study lacks visualization of training dynamics; plotting the evolution of semantic injection and suppression losses would help verify optimization stability and convergence.
- The paper misses key related works on image immunization, such as attention-based EditShield [1] and diffusion latent attack [2].

[1] Chen et. al. EditShield: Protecting Unauthorized Image Editing by Instruction-guided Diffusion Models, ECCV 2024
[2] Shih et. al. Pixel Is Not a Barrier: An Effective Evasion Attack for Pixel-Domain Diffusion Models, AAAI 2025

**Questions:**

- Could the authors include CLIPScore or Grounding DINO detection in the main results to evaluate semantic alignment, and provide additional metrics in the revision for completeness?
- When converting tensors back to images, clipping and quantization are applied. Could these operations break the attack by altering $\delta$ effective direction or strength and thus reduce the semantic injection/suppression effect?
- Could the two losses interfere or cancel each other out during optimization, given their opposite semantic objectives?
- Could the authors provide training curves of total, injection, and suppression losses to illustrate optimization stability and convergence?

I encourage the authors to strengthen the paper by addressing the weakness and  questions in the rebuttal.

---

> ### Author Response · Authors · 2025-11-23
>
> We sincerely appreciate the reviewer’s constructive feedback and insightful questions.
>
> > **W1**: The paper presents an empirical approach with limited theoretical justification.
>
> > **W2**: The authors do not provide a thorough discussion on why L_inj is effective in the cross-attention feature space or its theoretical justification, relying instead primarily on empirical validation.
>
> **A**: Thank you for the valuable feedback. In latent-diffusion models, cross-attention serves as the primary mechanism through which the conditioning prompt shapes the denoising trajectory. While the attention map $\mathcal{A}_l$ determines _where_ the model attends, the cross-attention output $CA_l$ is the quantity that directly updates the latent representation and therefore carries the actual semantic effect of the prompt.
>
> For this reason, our target semantic injection and source semantic suppression objectives operate on $CA_l$ rather than $\mathcal{A}_l$. The attention map is invariant to invertible transformations of the value vectors $V_l$, and thus cannot uniquely encode conditioning semantics, whereas modifying $CA_l$ directly influences how the model interprets the immunized image.
>
> We have provided a more detailed explanation in **Section 4.4** of the main manuscript to clarify why semantic injection and suppression must be applied at the cross-attention outputs. In addition, we include an ablation in Appendix Table 22 showing the results of applying the target semantic injection loss and source semantic suppression loss directly on the attention maps. We kindly invite the reviewer to refer to the revised section for the detailed justification.

---

> ### Author Response · Authors · 2025-11-23
>
> > **W3/Q1**: Could the authors include CLIPScore or Grounding DINO detection in the main results to evaluate semantic alignment, and provide additional metrics in the revision for completeness?
>
> **A**: Thank you for the constructive feedback. To evaluate how effectively our method prevents the model from relying on the source image during editing, we additionally measure the feature similarity between the source and edited images using CLIP and DINO embeddings, as suggested.
> These metrics directly quantify whether the edited output departs from the source semantics as intended. The results have been computed and updated in Tables 1-6 of the revised main manuscript. As shown in Tables 1–6, our method not only increases pixel-level distortion and perceptual dissimilarity but also effectively disrupts semantic consistency between the source and edited images.
>
>
> > **W4/Q3/Q4**: Could the two losses interfere or cancel each other out during optimization, given their opposite semantic objectives? Could the authors provide training curves of total, injection, and suppression losses to illustrate optimization stability and convergence?
>
> **A**: Thank you for your valuable feedback. To support the stability and convergence of our optimization, we have added the training curve visualization to Figure 20 in the Appendix. As training progresses, both the injection and suppression losses exhibit stable convergence behavior. Importantly, the two losses do not cancel each other out; rather, the semantic injection term reinforces the target semantics while the suppression term reduces the influence of the source semantics. These effects are complementary, resulting in stable optimization and improved protection performance when used together.

---

> ### Author Response · Authors · 2025-11-23
>
> > **W5**: The paper misses key related works on image immunization, such as attention-based EditShield [1] and diffusion latent attack [2].
>
>
> **A**: Thank you for pointing this out. We have added discussions on EditShield [1] and Pixel Is Not a Barrier [2] in the revised version (see Section 2). To ensure a fair and comprehensive comparison, we included the results of EditShield (ES) in the table (Table A below and Table 4 in our main manuscript), as an image-specific counterpart to our method. Please refer the corresponding black-box transferability results, provided in Appendix Table 8. In contrast, [2] targets PDM-based diffusion models, which differ from our setting, making a direct comparison not applicable; thus, we exclude it from our experiments.
>
> ### Table A: Comparison with image-specific methods (EA DA SA ES)  in the white-box setting on Stable Diffusion v1.5
> *Ours denotes the full method, while Ours\_DF uses only the target semantic injection loss in a data-free setting.
> Best results are in **bold**, second-best are $\underline{\text{underlined}}$.*
>
> |     |   |**Image-specific**   |   |    |     | **Universal**   | |
> |-|-|-|-|-|-|-|-|
> | Metric | Clean | EA | DA | SA | ES | Ours_DF | Ours |
> | **PSNR ↓** | -- | 14.75 ± 0.715 | 14.71 ± 0.951 | 16.28 ± 1.547 | 16.04 ± 0.820 | $\underline{14.68}$ ± 0.035 | **14.19** ± 0.059 |
> | **SSIM ↓** | -- | 0.386 ± 0.059 | 0.382 ± 0.082 | 0.431 ± 0.102 | $\underline{0.335}$ ± 0.067 | 0.378 ± 0.002 | **0.332** ± 0.003 |
> | **VIFp ↓** | -- | $\underline{0.088}$ ± 0.025 | 0.106 ± 0.037 | 0.138 ± 0.052 | 0.106 ± 0.031 | 0.106 ± 0.001 | **0.082** ± 0.001 |
> | **FSIM ↓** | -- | **0.637** ± 0.023 | 0.660 ± 0.037 | 0.714 ± 0.051 | 0.691 ± 0.028 | 0.666 ± 0.002 | $\underline{0.642}$ ± 0.003 |
> | **LPIPS ↑** | -- | $\underline{0.584}$ ± 0.034 | 0.552 ± 0.050 | 0.490 ± 0.064 | 0.553 ± 0.041 | 0.557 ± 0.002 | **0.606** ± 0.003 |
> | **Feat. Sim. (CLIP) ↓** | 0.744 | 0.677 ± 0.002 | **0.662** ± 0.002 | 0.705 ± 0.002 | 0.684 ± 0.003 | 0.685 ± 0.003 | $\underline{0.673}$ ± 0.002 |
> | **Feat. Sim. (DINO) ↓** | 0.534 | 0.373 ± 0.005 | **0.333** ± 0.006 | 0.447 ± 0.006 | 0.387 ± 0.004 | 0.376 ± 0.003 | $\underline{0.345}$ ± 0.002 |
>
> As shown in Table A, our method clearly surpasses EditShield across all metrics, while maintaining comparable results under the data-free setting, highlighting the superior generalization capability of our universal immunization approach.
>
> ---
> [1] Chen et. al,. EditShield: Protecting Unauthorized Image Editing by Instruction-guided Diffusion Models, ECCV 2024
>
> [2] Shih et. al., Pixel Is Not a Barrier: An Effective Evasion Attack for Pixel-Domain Diffusion Models, AAAI 2025

---

> ### Author Response · Authors · 2025-11-23
>
> > **Q2**: When converting tensors back to images, clipping and quantization are applied. Could these operations break the attack by altering  effective direction or strength and thus reduce the semantic injection/suppression effect?
>
> **A**: We apologize for the notational inconsistency in Algorithm 1, where the clipping step was omitted, and we have corrected this in the revised manuscript (Algorithm 1, Line 5).
>
> Our method employs a PGD-based optimization [3] where perturbations are clipped to the valid input range [0,1] at every iteration, as shown in the supplementary code at main.py L-151. As a result, the perturbation is inherently learned under the same clipping constraint applied at inference time, ensuring robustness to the range of real-world input images.
>
> Furthermore, we adopt a small update step size of 1/255, multiplying the sign of the gradient (±1) at each step. This prevents any quantization artifacts when converting tensors back to images. As a result, both clipping and quantization do not alter the effective direction or strength of the perturbation and thus do not affect the semantic injection or suppression effect.
>
> ---
> [3] Madry et. al., Towards Deep Learning Models Resistant to Adversarial Attacks, ICLR 2018

---

> ### Author Response · Authors · 2025-11-28
>
> We sincerely thank the reviewer for the constructive feedback and for raising the score. If there are any remaining concerns, please let us know. We are more than happy to address them.

---

### Official Review · Reviewer_nGCo · 2025-10-25

**Soundness:** 2
**Presentation:** 3
**Contribution:** 2
**Rating:** 4
**Confidence:** 3

**Summary:**

This paper proposes a universal image immunization framework that protects images from malicious diffusion-based editing by applying a single, broadly effective adversarial perturbation. Unlike image-specific defenses, the method generates a universal adversarial perturbation (UAP) that embeds a semantic target and suppresses original content, thereby misdirecting the diffusion model’s attention and preventing faithful or unauthorized semantic modifications.

**Strengths:**

- Research on anti-editing is meaningful and promising.
- The proposed universal adversarial perturbation (UAP) demonstrates greater effectiveness compared to prior per-image optimization approaches.
- Experimental results show that the proposed method achieves improved performance in several cases.

**Weaknesses:**

- During the editing phase, does the proposed method need to append the target prompt (e.g., “Ronaldo”) to the editing prompt? If so, how can it guarantee that a malicious user would use that specific prompt? If not, how does the UAP maintain robustness across different editing prompts, given that it appears to be trained with a fixed target prompt?

- How well does the UAP generalize to complex or lengthy editing prompts? Does its effectiveness degrade under more complicated prompt conditions?

- The UAP is trained on 10,000 random image–prompt pairs. How does the size of this training set influence the robustness and generalization of the learned perturbation?

- Since the primary goal is to defend against editing rather than to generate a target pattern, why is the target semantic injection loss necessary? Would using only the source semantic suppression loss suffice, and how would that affect performance?

**Questions:**

Please refer to the weakness part above.

---

> ### Author Response · Authors · 2025-11-23
>
> We sincerely appreciate the reviewer’s constructive feedback and insightful questions.
>
> > **W1**: During the editing phase, does the proposed method need to append the target prompt (e.g., “Ronaldo”) to the editing prompt? If so, how can it guarantee that a malicious user would use that specific prompt? If not, how does the UAP maintain robustness across different editing prompts, given that it appears to be trained with a fixed target prompt?
>
> **A**: We would like to clarify that our method does NOT require appending the target prompt to the editing prompt during the editing phase. A Universal adversarial perturbation (UAP) is generated once using the target prompt and then directly applied to the input image at inference time, without modifying the editing prompt provided by the user.
>
> To ensure robustness across different editing prompts, our method relies on two complementary objectives:
> 1) Target Semantic Injection, which encourages the UAP to encode the dominant semantics of the target concept in an image-agnostic manner, softly biasing the diffusion model toward the injected semantics; and
> 2) Source Semantic Suppression, which reduces the influence of generic content semantics internalized through large-scale training, making it difficult for the model to properly interpret the original image semantics.
>
> Together, these objectives disrupt the model’s ability to recognize the original content and steer the editing process toward the injected semantics (e.g., “tiger,” “Ronaldo”). This behavior is clearly demonstrated in Figure 2(c): when immunized, the attention induced by the target prompt aligns strongly with the injected target semantics, while the original content fails to capture meaningful attention. By weakening the model’s reliance on the original semantics in the input image, our UAP ensures strong protection regardless of the editing prompt.

---

> ### Author Response · Authors · 2025-11-23
>
> > **W2**: How well does the UAP generalize to complex or lengthy editing prompts? Does its effectiveness degrade under more complicated prompt conditions?
>
> **A**:
>
> We thank to the reviewer for this valuable suggestion. To further validate the effectiveness of our mode, we conducted an additional experiment using more complex and lengthy editing prompts. An example of the editing prompts used in this setting is: “A photo of a woman on a bright beach with smooth sand, soft waves, warm sunlight, light reflections, calm water, and an open coastal horizon.”
>
> ### Table A: White-box evaluation with complex editing prompts
> *Ours denotes the full method. The best result is in **bold**, and the second-best is $\underline{\text{underlined}}$.*
>
> | Metric | Clean | Enc. | Emb. | Map | Ours |
> |--------|-------|------|------|------|-------|
> | **PSNR ↓** | -- | 17.73 | $\underline{16.70}$ | 17.19 | **15.66** |
> | **SSIM ↓** | -- | 0.608 | $\underline{0.447}$ | 0.562 | **0.422** |
> | **VIFp ↓** | -- | 0.221 | $\underline{0.148}$ | 0.192 | **0.121** |
> | **FSIM ↓** | -- | 0.774 | $\underline{0.720}$ | 0.747 | **0.687** |
> | **LPIPS ↑** | -- | 0.369 | $\underline{0.511}$ | 0.417 | **0.548** |
> | **Feat. Sim. (CLIP) ↓** | 0.744 | 0.712 | $\underline{0.703}$ | 0.707 | **0.689** |
> | **Feat. Sim. (DINO) ↓** | 0.533 | 0.423 | 0.426 | $\underline{0.408}$ | **0.351** |
>
>
> As shown in Table A, our approach consistently achieves strong immunization performance compared to other universal baselines. These results demonstrate its effectiveness and further highlight its robustness against complicated editing prompts.

---

> ### Author Response · Authors · 2025-11-23
>
> > **W3**: The UAP is trained on 10,000 random image–prompt pairs. How does the size of this training set influence the robustness and generalization of the learned perturbation?
>
>
> **A**: Thank you for the suggestion. To examine the robustness and generalization with respect to the size of the training set, we conducted an additional ablation study by varying the number of training samples to 10, 100, and 1,000.
>
> ### Table B: Performance comparison across different numbers of training samples compared with image-specific methods.
> *Best results are in **bold**, second-best are $\underline{\text{underlined}}$.*
> |        |   |**Image-specific**   |   |      |       | **Universal**   | |  |        |     |
> |-|-|-|-|-|-|-|-|-|-|-|
> | Metric |Clean | EA | DA | SA | ES  | #10 | #100 | #1000 | #10000 | Ours_DF |
> | **PSNR ↓** | -- | 14.75 | 14.71 | 16.28 | 16.04  | 15.14 | 14.52 | $\underline{14.37}$ | **14.19** | 14.68|
> | **SSIM ↓** | -- | 0.386 | 0.382 | 0.431 | $\underline{0.335}$  | 0.408 | 0.373 | 0.359 | **0.332** | 0.378 |
> | **VIFp ↓** | -- | $\underline{0.088}$ | 0.106 | 0.138 | 0.106  | 0.124 | 0.100 | 0.093 | **0.082** | 0.106 |
> | **FSIM ↓** | -- | **0.637** | 0.660 | 0.714 | 0.691 | 0.681 | 0.658 | 0.652 | $\underline{0.642}$ | 0.666 |
> | **LPIPS ↑** | -- | $\underline{0.584}$ | 0.552 | 0.490 | 0.553  | 0.517 | 0.564 |0.575 | **0.606** | 0.557 |
> | **Feat. Sim. (CLIP) ↓** | 0.744 | 0.677 | **0.662** | 0.705 | 0.684| 0.696 | 0.686 | 0.682 | $\underline{0.673}$ |0.685 |
> | **Feat. Sim. (DINO) ↓** | 0.534 | 0.373 | **0.333** | 0.447 | 0.387 | 0.402 | 0.365 | 0.360 | $\underline{0.345}$ | 0.376 | $\underline{0.345}$ |
>
>
> As shown in Table B, although the performance steadily improves as the number of training samples increases, the overall performance differences remain **relatively small**, demonstrating that our method is robust even when only a few training samples are available. Interestingly, our data-free UAP even outperforms the model trained with only 10 samples, likely due to overfitting—very limited data causes the model to rely on instance-specific semantics rather than learning generalizable protection.
> Moreover, compared to image-specific methods—which have a significantly stronger advantage because they directly utilize the target image to tailor the perturbation—our universal UAP still performs favorably. Specifically, with only 10 samples, our method outperforms the Semantic Attack (SA), and with 100 samples, it even surpasses both the Encoder Attack (EA) and Diffusion Attack (DA) methods.

---

> ### Author Response · Authors · 2025-11-23
>
> > **W4**: Since the primary goal is to defend against editing rather than to generate a target pattern, why is the target semantic injection loss necessary? Would using only the source semantic suppression loss suffice, and how would that affect performance?
>
>
>
> **A**:
> We would like to clarify that the primary goal of our method is to inject arbitrary target semantics into the image so that diffusion models are misled during editing, thereby protecting against malicious manipulation. While our Source Semantic Suppression Loss provides benefits (Table 5 in our main manuscript), it alone is insufficient for strong protection in our universal, image-agnostic setting. Because the UAP is trained without access to the protected image, it can only suppress generic content semantics, not remove instance-specific ones.
> In contrast, the Target Semantic Injection Loss plays a crucial role by embedding the target semantics directly into the image representation, leading the diffusion model to focus more on the injected semantics than the original semantics during editing.
>
> This is strongly validated by the ablation in Table C: removing the injection loss results in a clear performance drop, confirming that suppression alone is insufficient for robust protection. Notably, suppression becomes most effective only when combined with injection, confirming that the two losses work in a complementary manner.
> ### Table C: Ablation study for loss functions
> *Best results are in **bold**, second-best are $\underline{\text{underlined}}$.*
>
> | Metric | Clean | $\mathcal{L}_\text{sup}$ | $\mathcal{L}_\text{inj}$ |$\mathcal{L}_\text{inj+sup}$ |
> |-------|------|------------------------|------------------------|----------------------------------------------|
> | **PSNR ↓** | -- | 16.11 | $\underline{14.41}$ | **14.19** |
> | **SSIM ↓** | -- | 0.420 | $\underline{0.367}$ | **0.332** |
> | **VIFp ↓** | -- | 0.140 | $\underline{0.096}$ | **0.082** |
> | **FSIM ↓** | -- | 0.697 | $\underline{0.655}$ | **0.642** |
> | **LPIPS ↑** | -- | 0.490 | $\underline{0.585}$ | **0.606** |
> | **Feat. Sim. (CLIP) ↓** | 0.744 | 0.688 | $\underline{0.680}$ | **0.673** |
> | **Feat. Sim. (DINO) ↓** | 0.534 | 0.415 | $\underline{0.372}$ | **0.345** |

---

> ### Author Response · Authors · 2025-11-29
>
> Dear reviewer nGCo,
>
> We carefully provide visualization to further strengthen our statement on _“how the UAP maintains robustness across different editing prompts despite being trained with a fixed target prompt”_ (**W1**), illustrating how the model responds to the injected target semantics. The result is provided in Figure 24 in the appendix, and we would greatly appreciate it if you could take a look.
>
> Specifically, we feed the immunized latent into the diffusion model using the same noise schedule as in editing (i.e., strength = 0.8), but with a null prompt, so that no text guidance is provided. Remarkably, while the original lion images produce only stylistically similar but entirely unrelated outputs under null-prompt generation, our method consistently generates results that remain highly aligned with the target image, even in the absence of any textual guidance.
>
> This indicates that the UAP acts as a target-driven guidance signal at the latent level, allowing the model to preserve the target semantics even when the target prompt is absent during inference. Consequently, this supports why our method remains robust across diverse editing prompts: the UAP embeds target semantics directly into the latent, enabling effective immunization independent of the editing prompt.

---

### Author Response · Authors · 2025-11-23

Dear reviewers,

We sincerely appreciate the time and effort you have invested in reviewing our paper. Your thoughtful and constructive feedback has been immensely helpful in strengthening and refining our work.

As reviewers highlighted, our work proposes a significant universal immunization approach(nGCo) with a well-structured dual-loss design (8CoU, Hbgg, S8F9), effectively surpasses universal baselines and image-specific methods(nGCo, 8CoU, S8F9), produces strong scalability and practicality even in a data-free setting (8CoU, S8F9), and is clearly written with intuitive presentation (S8F9, Hbgg).

In response to your feedback, we have conducted additional experiments and incorporated the findings into the revised draft. Below is an outline of the key updates:

* Evaluation of the effectiveness of Universal Image Immunization against long and complex editing prompt (nGCo)

* Ablation studies for the number of training samples (nGCo)

* Effect of optimizing UAP only with source semantic suppression loss (nGCo, S8F9)

* Clear justification for utilizing cross-attention output for target semantic injection and source semantic suppression loss (8CoU, S8F9)

* Visualization of training curve for two loss components (8CoU)

* Ablation studies for different target identity (S8F9)

* Evaluation of the generalizability of source semantic suppression loss (S8F9)

* Evaluation of the immunization performance trained with grid pattern-like target image (Hbgg)

* Evaluation of the effectiveness of our UAP  under image-specific perturbation budget (Hbgg)

We believe that Universal Image Immunization can be a useful addition to the ICLR community, guided by your valuable feedback helping us enhance the clarity and depth of our work.

Thank you once again for your constructive reviews and support.

Authors

---

### Author Response · Authors · 2025-12-03

Dear AC,

Thank you for your time and effort in evaluating our paper. During the rebuttal period, we sincerely provided extensive experiments and clarifications to directly address the reviewers’ concerns. Below, we summarize the key points.

---
**(1) Strengths acknowledged by reviewers**
* Our framework is well motivated (Hbgg), simple yet effective (8CoU), and introduces well-structured dual-loss design (8CoU, S8F9, Hbgg) to address the significant universal immunization approach (nGCo).
* Our method surpasses both universal baselines and image-specific methods(nGCo, 8CoU, S8F9), while maintaining visual quality and robustness (8CoU).
* Strong scalability and practicality are demonstrated (S8F9, Hbgg), even in a data-free setting (8CoU, Hbgg) and across different diffusion pipelines (8CoU, S8F9, Hbgg).
* The paper is clearly written with intuitive presentation (S8F9, Hbgg).

We sincerely appreciate these positive and encouraging assessments.

---
**(2) Main concerns and our responses**
* **C1: Reliance on the target prompt during editing; limited generality under arbitrary editing prompts (nGCo)**

  We clarified that our method does NOT require the target prompt (e.g., “Ronaldo”) during editing. Additional null-prompt experiments (Appendix Fig. 24) show that injected semantics remain dominant, verifying strong robustness of our UAP under arbitrary editing prompts.
* **C2: Justification for operating on the cross-attention outputs; comparison with applying the proposed losses to attention map instead (8CoU, nGCo, S8F9)**

  We expanded the theoretical justification in Section 4.4 and clarified why cross-attention outputs are the most suitable space for semantic injection and suppression. We also added comparisons with an attention-map baseline (Appendix Tab. 21), showing that operating on cross-attention outputs yields clearer semantic signals and superior performance.
* **C3: Need for semantic-level evaluations (8CoU)**

  We added CLIP/DINO-based semantic similarity metrics (Tab. 1–6), reinforcing the semantic effectiveness of our immunization.
* **C4: Fairness of comparison with Semantic Attack (Hbgg)**

  We clarified that Semantic Attack inherently uses an ℓ∞ projection in its original formulation, and our implementation follows the same constraint, ensuring a fair comparison.

---
**_Additional experiments_**

To strengthen completeness and address reviewer concerns, we performed the following additional evaluations:
* Robustness under complex prompts  Appendix Tab. 18 (nGCo)
* Ablation on the number of training samples: Appendix Tab. 19 (nGCo)
* Comparison with an additional image-specific method: Tab. 4, 8 (8CoU)
* Training-curve visualization to examine potential loss interference: Appendix Fig. 20 (8CoU)
* Robustness of our source semantic suppression loss to unseen semantics: Appendix Tab. 22 (S8F9)
* Evaluation of our UAP trained with geometric target image (e.g. checkerboard): Appendix Tab. 23 (Hbgg)
* Comparison under the same perturbation budget used by image-specific methods: Appendix Tab. 24, 25 (Hbgg)


**_Additional Interesting Question from Reviewer S8F9_**

Reviewer S8F9 requested a deeper assessment of how strongly identity-level semantics are injected. In response, we added ablations examining the transfer of attributes such as gender, age, and skin tone (Fig. 21 and 23), as well as cross-category effects by applying a human-trained UAP to animal images (Fig. 22). These results provide a detailed analysis directly aligned with the reviewer’s request.

---
**(3) Reviewer-specific follow-ups**

During the rebuttal, we made sincere efforts to continue the discussion with the reviewers.
* Reviewer S8F9 expressed that most concerns were resolved and engaged in deeper discussion on semantic injection. We also addressed the effectiveness on cross-attention outputs, robustness to training samples, and real-image evaluation (already included in the Appendix).
* Reviewer 8CoU did not leave further comments, but the reviewer had already raised the score earlier in the process (4 to 6), which we interpret as a positive signal that our clarifications, expanded theoretical justification, added CLIP/DINO metrics, training-curve visualization and the inclusion of missing references with comparison results successfully addressed the major concerns.

Although we have not yet heard back from Reviewers nCGo and Hbgg, we addressed their concerns thoroughly:
* For Reviewer nCGo, we clarified that the target prompt is not required during editing. Null-prompt experiments confirm robustness to arbitrary prompts, and additional evaluations further show generalization to complex prompts, robustness to limited training data, and the necessity of semantic injection.
* For Reviewer Hbgg, we clarified that Semantic Attack does indeed employ a perturbation constraint and that our comparison is conducted under matched budgets, fully addressing the reviewer’s only identified weakness.

Sincerely,

4421 authors

---

### Meta-Review · Area_Chair_cy2f · 2026-01-09

**Summary:**

The paper applies universal adversarial attack to image protection framework, so that the optimized perturbation can protect images without additional cost. Reviewer scores were initially biased a bit towards rejection (4, 4, 4, 8). In the rebuttal, the authors have addressed many of the reviewers’ concerns with additional experiments; additional baselines, complex prompts, training sample size, etc. However, AC finds several major concerns raised by nGCo are valid yet still outstanding, particularly regarding robustness to prompt variations and complex prompts. Given the threat scenarios targeted by the paper, it is important to demonstrate that the proposed method is sufficiently robust to prompt changes by attackers, which is not yet extensively validated beyond the newly added Table 18 in the current version. AC also believes that the robustness evaluation in Table 3 should be strengthened by including image-specific baseline methods.

**Reviewer Concerns:**

Some of the major concerns remain outstanding, e.g., robustness to prompt variations and complex prompts, and AC agrees on these concerns.

**Reviewer Scores:**

- Reviewer nGCo: Initially 4. Would maintain the original score.
- Reviewer 8CoU: Initially 4. The score was increased to 6 before score resetting.
- Reviewer S8F9: Initially 4. Would likely increase to 6.
- Reviewer Hbgg: Initially 8. Would maintain the original score.

---

### Decision · Program_Chairs · 2026-01-26

Reject